# Self-Supervised Learning with Lie Symmetries for Partial Differential Equations

**Grégoire Mialon**[†]
Meta, FAIR

**Quentin Garrido**[†]
Meta, FAIR
Univ Gustave Eiffel, CNRS, LIGM

**Hannah Lawrence**
Meta, FAIR
MIT

**Danyal Rehman**
MIT

**Yann LeCun**
Meta, FAIR
NYU

**Bobak T. Kiani**[*]
MIT

## Abstract

Machine learning for differential equations paves the way for computationally efficient alternatives to numerical solvers, with potentially broad impacts in science and engineering. Though current algorithms typically require simulated training data tailored to a given setting, one may instead wish to learn useful information from heterogeneous sources, or from real dynamical systems observations that are messy or incomplete. In this work, we learn general-purpose representations of PDEs from heterogeneous data by implementing joint embedding methods for self-supervised learning (SSL), a framework for unsupervised representation learning that has had notable success in computer vision. Our representation outperforms baseline approaches to invariant tasks, such as regressing the coefficients of a PDE, while also improving the time-stepping performance of neural solvers. We hope that our proposed methodology will prove useful in the eventual development of general-purpose foundation models for PDEs.

## 1 Introduction

Dynamical systems governed by differential equations are ubiquitous in fluid dynamics, chemistry, astrophysics, and beyond. Accurately analyzing and predicting the evolution of such systems is of paramount importance, inspiring decades of innovation in algorithms for numerical methods. However, high-accuracy solvers are often computationally expensive. Machine learning has recently arisen as an alternative method for analyzing differential equations at a fraction of the cost [1, 2, 3]. Typically, the neural network for a given equation is trained on simulations of that same equation, generated by numerical solvers that are high-accuracy but comparatively slow [4]. What if we instead wish to learn from heterogeneous data, e.g., data with missing information, or gathered from actual observation of varied physical systems rather than clean simulations?

For example, we may have access to a dataset of instances of time-evolution, stemming from a family of partial differential equations (PDEs) for which important characteristics of the problem, such as viscosity or initial conditions, vary or are unknown. In this case, representations learned from such a large, "unlabeled" dataset could still prove useful in learning to identify unknown characteristics, given only a small dataset "labeled" with viscosities or reaction constants. Alternatively, the "unlabeled" dataset may contain evolutions over very short periods of time, or with missing time intervals; possible goals are then to learn representations that could be useful in filling in these gaps, or regressing other quantities of interest.

---

[*]Correspondence to: gmialon@meta.com, garridoq@meta.com, and bkiani@mit.edu, [†] Equal contribution

37th Conference on Neural Information Processing Systems (NeurIPS 2023).

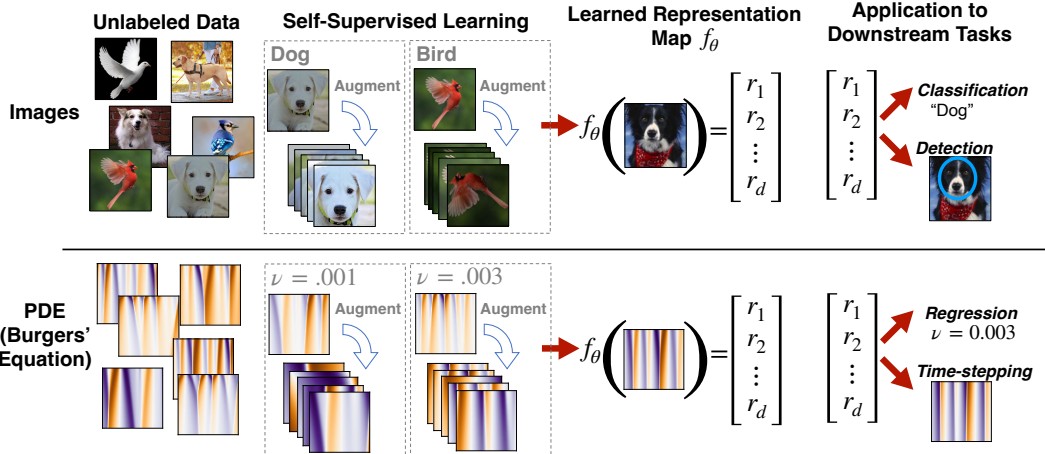

Figure 1: A high-level overview of the self-supervised learning pipeline, in the conventional setting of image data (top row) as well as our proposed setting of a PDE (bottom row). Given a large pool of unlabeled data, self-supervised learning uses augmentations (e.g. color-shifting for images, or Lie symmetries for PDEs) to train a network $f_\theta$ to produce useful representations from input images. Given a smaller set of labeled data, these representations can then be used as inputs to a supervised learning pipeline, performing tasks such as predicting class labels (images) or regressing the kinematic viscosity $\nu$ (Burgers' equation). Trainable steps are shown with red arrows; importantly, the representation function learned via SSL is not altered during application to downstream tasks.

To tackle these broader challenges, we take inspiration from the recent success of self-supervised learning (SSL) as a tool for learning rich representations from large, unlabeled datasets of text and images [5, 6]. Building such representations from and for scientific data is a natural next step in the development of machine learning for science [7]. In the context of PDEs, this corresponds to learning representations from a large dataset of PDE realizations "unlabeled" with key information (such as kinematic viscosity for Burgers' equation), before applying these representations to solve downstream tasks with a limited amount of data (such as kinematic viscosity regression), as illustrated in Figure 1.

To do so, we leverage the joint embedding framework [8] for self-supervised learning, a popular paradigm for learning visual representations from unlabeled data [9, 10]. It consists of training an encoder to enforce similarity between embeddings of two augmented versions of a given sample to form useful representations. This is guided by the principle that representations suited to downstream tasks (such as image classification) should preserve the common information between the two augmented views. For example, changing the color of an image of a dog still preserves its semantic meaning and we thus want similar embeddings under this augmentation. Hence, the choice of augmentations is crucial. For visual data, SSL relies on human intuition to build hand-crafted augmentations (e.g. recoloring and cropping), whereas PDEs are endowed with a group of symmetries preserving the governing equations of the PDE [11, 12]. These symmetry groups are important because creating embeddings that are invariant under them would allow to capture the underlying dynamics of the PDE. For example, solutions to certain PDEs with periodic boundary conditions remain valid solutions after translations in time and space. There exist more elaborate equation-specific transformations as well, such as Galilean boosts and dilations (see Appendix E). Symmetry groups are well-studied for common PDE families, and can be derived systematically or calculated from computer algebra systems via tools from Lie theory [11, 13, 14].

**Contributions:** We present a general framework for performing SSL for PDEs using their corresponding symmetry groups. In particular, we show that by exploiting the analytic group transformations from one PDE solution to another, we can use joint embedding methods to generate useful representations from large, heterogeneous PDE datasets. We demonstrate the broad utility of these representations on downstream tasks, including regressing key parameters and time-stepping, on simulated physically-motivated datasets. Our approach is applicable to any family of PDEs, harnesses the well-understood mathematical structure of the equations governing PDE data — a luxury not typically available in non-scientific domains — and demonstrates more broadly the promise of

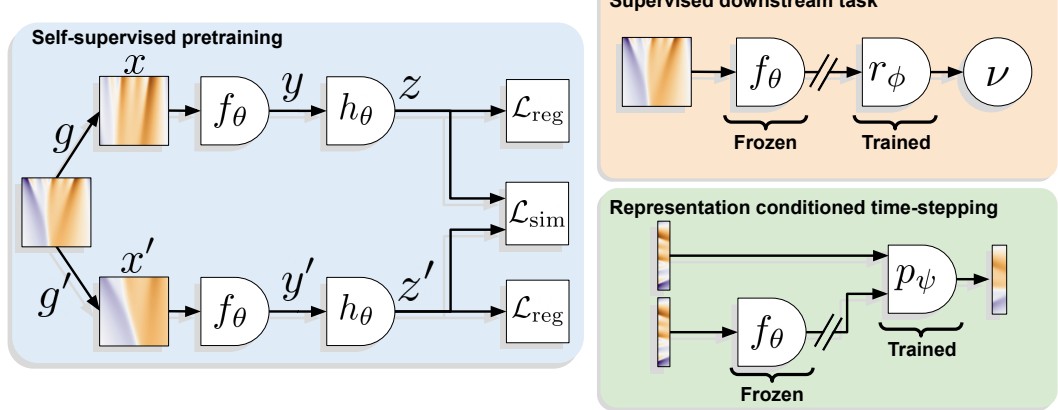

Figure 2: Pretraining and evaluation frameworks, illustrated on Burgers' equation. **(Left)** Self-supervised pretraining. We generate augmented solutions $x$ and $x'$ using Lie symmetries parametrized by $g$ and $g'$ before passing them through an encoder $f_\theta$, yielding representations $y$. The representations are then input to a projection head $h_\theta$, yielding embeddings $z$, on which the SSL loss is applied. **(Right)** Evaluation protocols for our pretrained representations $y$. On new data, we use the computed representations to either predict characteristics of interest, or to condition a neural network or operator to improve time-stepping performance.

adapting self-supervision to the physical sciences. We hope this work will serve as a starting point for developing foundation models on more complex dynamical systems using our framework.

## 2 Methodology

We now describe our general framework for learning representations from and for diverse sources of PDE data, which can subsequently be used for a wide range of tasks, ranging from regressing characteristics of interest of a PDE sample to improving neural solvers. To this end, we adapt a popular paradigm for representation learning without labels: the joint-embedding self-supervised learning.

### 2.1 Self-Supervised Learning (SSL)

**Background:** In the joint-embedding framework, input data is transformed into two separate "views", using augmentations that preserve the underlying information in the data. The augmented views are then fed through a learnable encoder, $f_\theta$, producing representations that can be used for downstream tasks. The SSL loss function is comprised of a similarity loss $\mathcal{L}_{\text{sim}}$ between projections (through a projector $h_\theta$, which helps generalization [15]) of the pairs of views, to make their representations invariant to augmentations, and a regularization loss $\mathcal{L}_{\text{reg}}$, to avoid trivial solutions (such as mapping all inputs to the same representation). The regularization term can consist of a repulsive force between points, or regularization on the covariance matrix of the embeddings. Both function similarly, as shown in [16]. This pretraining procedure is illustrated in Fig. 2 (left) in the context of Burgers' equation.

In this work, we choose variance-invariance-covariance regularization (VICReg) as our self-supervised loss function [9]. Concretely, let $\boldsymbol{Z}, \boldsymbol{Z}' \in \mathbb{R}^{N \times D}$ contain the $D$-dimensional representations of two batches of $N$ inputs with $D \times D$ centered covariance matrices, $\mathrm{Cov}(\boldsymbol{Z})$ and $\mathrm{Cov}(\boldsymbol{Z}')$. Rows $\boldsymbol{Z}_{i,:}$ and $\boldsymbol{Z}'_{i,:}$ are two views of a shared input. The loss over this batch includes a term to enforce similarity ($\mathcal{L}_{\text{sim}}$) and a term to avoid collapse and regularize representations ($\mathcal{L}_{\text{reg}}$) by pushing elements of the encodings to be statistically identical:

$$\mathcal{L}(\boldsymbol{Z}, \boldsymbol{Z}') \approx \frac{\lambda_{inv}}{N} \underbrace{\sum_{i=1}^{N} \|\boldsymbol{Z}_{i,:} - \boldsymbol{Z}'_{i,:}\|_2^2}_{\mathcal{L}_{\text{sim}}(\boldsymbol{Z},\boldsymbol{Z}')} + \frac{\lambda_{reg}}{D} \underbrace{(\| \mathrm{Cov}(\boldsymbol{Z}) - \boldsymbol{I}\|_F^2 + \| \mathrm{Cov}(\boldsymbol{Z}') - \boldsymbol{I}\|_F^2)}_{\mathcal{L}_{\text{reg}}(\boldsymbol{Z})+\mathcal{L}_{\text{reg}}(\boldsymbol{Z}')}, \quad (1)$$

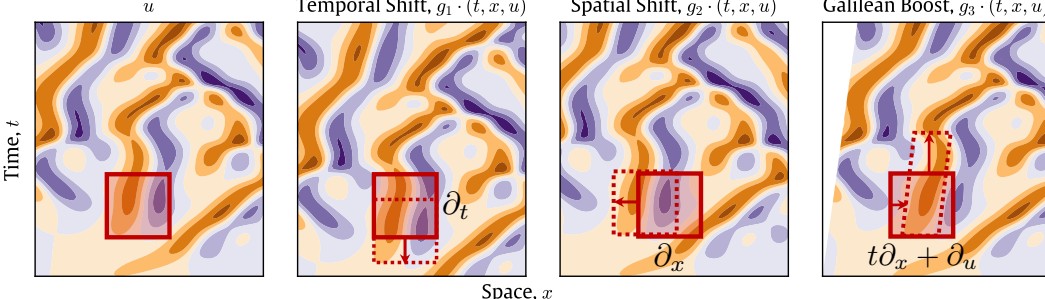

Figure 3: One parameter Lie point symmetries for the Kuramoto-Sivashinsky (KS) PDE. The transformations (left to right) include the un-modified solution ($u$), temporal shifts ($g_1$), spatial shifts ($g_2$), and Galilean boosts ($g_3$) with their corresponding infinitesimal transformations in the Lie algebra placed inside the figure. The shaded red square denotes the original $(x, t)$, while the dotted line represents the same points after the augmentation is applied.

where $\| \cdot \|_F$ denotes the matrix Frobenius norm and $\lambda_{inv}, \lambda_{reg} \in \mathbb{R}^+$ are hyperparameters to weight the two terms. In practice, VICReg separates the regularization $\mathcal{L}_{reg}(\boldsymbol{Z})$ into two components to handle diagonal and non-diagonal entries $\mathrm{Cov}(\boldsymbol{Z})$ separately. For full details, see Appendix C.

**Adapting VICReg to learn from PDE data:**  Numerical PDE solutions typically come in the form of a tensor of values, along with corresponding spatial and temporal grids. By treating the spatial and temporal information as supplementary channels, we can use existing methods developed for learning image representations. As an illustration, a numerical solution to Burgers consists of a velocity tensor with shape $(t, x)$: a vector of $t$ time values, and a vector of $x$ spatial values. We therefore process the sample to obtain a $(3, t, x)$ tensor with the last two channels encoding spatial and temporal discretization, which can be naturally fed to neural networks tailored for images such as ResNets [17]. From these, we extract the representation before the classification layer (which is unused here). It is worth noting that convolutional neural networks have become ubiquitous in the literature [18, 12]. While the VICReg default hyper-parameters did not require substantial tuning, tuning was crucial to probe the quality of our learned representations to monitor the quality of the pre-training step. Indeed, SSL loss values are generally not predictive of the quality of the representation, and thus must be complemented by an evaluation task. In computer vision, this is done by freezing the encoder, and using the features to train a linear classifier on ImageNet. In our framework, we pick regression of a PDE coefficient, or regression of the initial conditions when there is no coefficient in the equation. The latter, commonly referred to as the inverse problem, has the advantage of being applicable to any PDE, and is often a challenging problem in the numerical methods community given the ill-posed nature of the problem [19]. Our approach for a particular task, kinematic viscosity regression, is schematically illustrated in Fig. 2 (top right). More details on evaluation tasks are provided in Section 4.

## 2.2 Augmentations and PDE Symmetry Groups

**Background:**  PDEs formally define a systems of equations which depend on derivatives of input variables. Given input space $\Omega$ and output space $\mathcal{U}$, a PDE $\Delta$ is a system of equations in independent variables $\boldsymbol{x} \in \Omega$, dependent variables $\boldsymbol{u} : \Omega \to \mathcal{U}$, and derivatives $(\boldsymbol{u_x}, \boldsymbol{u_{xx}}, \dots)$ of $\boldsymbol{u}$ with respect to $\boldsymbol{x}$. For example, the Kuramoto–Sivashinsky equation is given by

$$\Delta(x, t, u) = u_t + uu_x + u_{xx} + u_{xxxx} = 0. \tag{2}$$

Informally, a symmetry group of a PDE $G$ [2] acts on the total space via smooth maps $G : \Omega \times \mathcal{U} \to \Omega \times \mathcal{U}$ taking solutions of $\Delta$ to other solutions of $\Delta$. More explicitly, $G$ is contained in the symmetry group of $\Delta$ if outputs of group operations acting on solutions are still a solution of the PDE:

$$\Delta(\boldsymbol{x}, \boldsymbol{u}) = 0 \implies \Delta[g \cdot (\boldsymbol{x}, \boldsymbol{u})] = 0, \quad \forall g \in G. \tag{3}$$

---

[2]A group $G$ is a set closed under an associative binary operation containing an identity element $e$ and inverses (*i.e.*, $e \in G$ and $\forall g \in G : g^{-1} \in G$). $G : \mathcal{X} \to \mathcal{X}$ acts on a space $\mathcal{X}$ if $\forall x \in \mathcal{X}, \forall g, h \in G : ex = x$ and $(gh)x = g(hx)$.

For PDEs, these symmetry groups can be analytically derived [11] (see also Appendix A for more formal details). The types of symmetries we consider are so-called Lie point symmetries $g : \Omega \times \mathcal{U} \to \Omega \times \mathcal{U}$, which act smoothly at any given point in the total space $\Omega \times \mathcal{U}$. For the Kuramoto-Sivashinsky PDE, these symmetries take the form depicted in Fig. 3:

$$
\begin{aligned}
\text{Temporal Shift:} \quad & g_1(\epsilon) : (x, t, u) \mapsto (x, t + \epsilon, u) \\
\text{Spatial Shift:} \quad & g_2(\epsilon) : (x, t, u) \mapsto (x + \epsilon, t, u) \\
\text{Galilean Boost:} \quad & g_3(\epsilon) : (x, t, u) \mapsto (x + \epsilon t, t, u + \epsilon)
\end{aligned}
\tag{4}
$$

As in this example, every Lie point transformation can be written as a one parameter transform of $\epsilon \in \mathbb{R}$ where the transformation at $\epsilon = 0$ recovers the identity map and the magnitude of $\epsilon$ corresponds to the "strength" of the corresponding augmentation.[3] Taking the derivative of the transformation at $\epsilon = 0$ with respect to the set of all group transformations recovers the Lie algebra of the group (see Appendix A). Lie algebras are vector spaces with elegant properties (e.g., smooth transformations can be uniquely and exhaustively implemented), so we parameterize augmentations in the Lie algebra and implement the corresponding group operation via the exponential map from the algebra to the group. Details are contained in Appendix B.

**PDE symmetry groups as SSL augmentations, and associated challenges:** Symmetry groups of PDEs offer a technically sound basis for the implementation of augmentations; nevertheless, without proper considerations and careful tuning, SSL can fail to work successfully [20]. Although we find the marriage of these PDE symmetries with SSL quite natural, there are several subtleties to the problem that make this task challenging. Consistent with the image setting, we find that, among the list of possible augmentations, crops are typically the most effective of the augmentations in building useful representations [21]. Selecting a sensible subset of PDE symmetries requires some care; for example, if one has a particular invariant task in mind (such as regressing viscosity), the Lie symmetries used should neither depend on viscosity nor change the viscosity of the output solution. Morever, there is no guarantee as to which Lie symmetries are the most "natural", *i.e.* most likely to produce solutions that are close to the original data distribution; this is also likely a confounding factor when evaluating their performance. Finally, precise derivations of Lie point symmetries require knowing the governing equation, though a subset of symmetries can usually be derived without knowing the exact form of the equation, as certain families of PDEs share Lie point symmetries and many symmetries arise from physical principles and conservation laws.

**Sampling symmetries:** We parameterize and sample from Lie point symmetries in the Lie algebra of the group, to ensure smoothness and universality of resulting maps in some small region around the identity. We use Trotter approximations of the exponential map, which are efficiently tunable to small errors, to apply the corresponding group operation to an element in the Lie algebra (see Appendix B) [22, 23]. In our experiments, we find that Lie point augmentations applied at relatively small strengths perform the best (see Appendix E), as they are enough to create informative distortions of the input when combined. Finally, boundary conditions further complicate the simplified picture of PDE symmetries, and from a practical perspective, many of the symmetry groups (such as the Galilean Boost in Fig. 3) require a careful rediscretization back to a regular grid of sampled points.

## 3 Related Work

In this section, we provide a concise summary of research related to our work, reserving Appendix D for more details. Our study derives inspiration from applications of Self-Supervised Learning (SSL) in building pre-trained foundational models [24]. For physical data, models pre-trained with SSL have been implemented in areas such as weather and climate prediction [7] and protein tasks [25, 26], but none have previously used the Lie symmetries of the underlying system. The SSL techniques we use are inspired by similar techniques used in image and video analysis [9, 20], with the hopes of learning rich representations that can be used for diverse downstream tasks.

Symmetry groups of PDEs have a rich history of study [11, 13]. Most related to our work, [12] used Lie point symmetries of PDEs as a tool for augmenting PDE datasets in supervised tasks. For some PDEs, previous works have explicitly enforced symmetries or conservation laws by for example constructing networks equivariant to symmetries of the Navier Stokes equation [27], parameterizing

---

[3]Technically, $\epsilon$ is the magnitude and direction of the transformation vector for the basis element of the corresponding generator in the Lie algebra.

networks to satisfy a continuity equation [28], or enforcing physical constraints in dynamic mode decomposition [29]. For Hamiltonian systems, various works have designed algorithms that respect the symplectic structure or conservation laws of the Hamiltonian [30, 31].

## 4  Experiments

**Equations considered:**  We focus on flow-related equations here as a testing ground for our methodology. In our experiments, we consider the four equations below, which are 1D evolution equations apart from the Navier-Stokes equation, which we consider in its 2D spatial form. For the 1D flow-related equations, we impose periodic boundary conditions with $\Omega = [0, L] \times [0, T]$. For Navier-Stokes, boundary conditions are Dirichlet ($v = 0$) as in [18]. Symmetries for all equations are listed in Appendix E.

1. The **viscous Burgers' Equation**, written in its "standard" form, is a nonlinear model of dissipative flow given by
$$u_t + uu_x - \nu u_{xx} = 0, \tag{5}$$
    where $u(x, t)$ is the velocity and $\nu \in \mathbb{R}^+$ is the kinematic viscosity.

2. The **Korteweg-de Vries (KdV)** equation models waves on shallow water surfaces as
$$u_t + uu_x + u_{xxx} = 0, \tag{6}$$
    where $u(x, t)$ represents the wave amplitude.

3. The **Kuramoto-Sivashinsky (KS)** equation is a model of chaotic flow given by
$$u_t + uu_x + u_{xx} + u_{xxxx} = 0, \tag{7}$$
    where $u(x, t)$ is the dependent variable. The equation often shows up in reaction-diffusion systems, as well as flame propagation problems.

4. The **incompressible Navier-Stokes** equation in two spatial dimensions is given by
$$\boldsymbol{u}_t = -\boldsymbol{u} \cdot \nabla \boldsymbol{u} - \frac{1}{\rho}\nabla p + \nu \nabla^2 \boldsymbol{u} + \boldsymbol{f}, \quad \nabla \boldsymbol{u} = 0, \tag{8}$$
    where $\boldsymbol{u}(\boldsymbol{x}, t)$ is the velocity vector, $p(\boldsymbol{x}, t)$ is the pressure, $\rho$ is the fluid density, $\nu$ is the kinematic viscosity, and $\boldsymbol{f}$ is an external added force (buoyancy force) that we aim to regress in our experiments.

Solution realizations are generated from analytical solutions in the case of Burgers' equation or pseudo-spectral methods used to generate PDE learning benchmarking data (see Appendix F) [12, 18, 32]. Burgers', KdV and KS's solutions are generated following the process of [12] while for Navier Stokes we use the conditioning dataset from [18]. The respective characteristics of our datasets can be found in Table 1.

**Pretraining:**  For each equation, we pretrain a ResNet18 with our SSL framework for 100 epochs using AdamW [33], a batch size of 32 (64 for Navier-Stokes) and a learning rate of 3e-4. We then freeze its weights. To evaluate the resulting representation, we (i) train a linear head on top of our features and on a new set of labeled realizations, and (ii) condition neural networks for time-stepping on our representation. Note that our encoder learns from heterogeneous data in the sense that for a given equation, we grouped time evolutions with different parameters and initial conditions.

### 4.1  Equation parameter regression

We consider the task of regressing equation-related coefficients in Burgers' equation and the Navier-Stokes' equation from solutions to those PDEs. For KS and KdV we consider the inverse probem of regressing initial conditions. We train a linear model on top of the pretrained representation for the downstream regression task. For the baseline supervised model, we train the same architecture, *i.e.* a ResNet18, using the MSE loss on downstream labels. Unless stated otherwise, we train the linear model for 30 epochs using Adam. Further details are in Appendix F.

**Kinematic viscosity regression (Burgers):** We pretrain a ResNet18 on $10,000$ unlabeled realizations of Burgers' equation, and use the resulting features to train a linear model on a smaller, labeled

Table 1: Downstream evaluation of our learned representations for four classical PDEs (averaged over three runs, the lower the better ($\downarrow$)). The normalized mean squared error (NMSE) over a batch of $N$ outputs $\widehat{\boldsymbol{u}}_k$ and targets $\boldsymbol{u}_k$ is equal to $\text{NMSE} = \frac{1}{N}\sum_{k=1}^{N}\|\widehat{\boldsymbol{u}}_k - \boldsymbol{u}_k\|_2^2/\|\widehat{\boldsymbol{u}}_k\|_2^2$. Relative error is similarly defined as $\text{RE} = \frac{1}{N}\sum_{k=1}^{N}\|\widehat{\boldsymbol{u}}_k - \boldsymbol{u}_k\|_1/\|\widehat{\boldsymbol{u}}_k\|_1$ For regression tasks, the reported errors with supervised methods are the best performance across runs with Lie symmetry augmentations applied. For timestepping, we report NMSE for KdV, KS and Burgers as in [12], and MSE for Navier-Stokes for comparison with [18].

| Equation | KdV | KS | Burgers | Navier-Stokes |
|---|---|---|---|---|
| SSL dataset size | 10,000 | 10,000 | 10,000 | 26,624 |
| Sample format $(t, x, (y))$ | 256×128 | 256×128 | 448×224 | 56×128×128 |
| Characteristic of interest | Init. coeffs | Init. coeffs | Kinematic viscosity | Buoyancy |
| Regression metric | NMSE ($\downarrow$) | NMSE ($\downarrow$) | Relative error %($\downarrow$) | MSE ($\downarrow$) |
| Supervised | $0.102 \pm 0.007$ | $0.117 \pm 0.009$ | $1.18 \pm 0.07$ | $0.0078 \pm 0.0018$ |
| SSL repr. + linear head | $\mathbf{0.033 \pm 0.004}$ | $\mathbf{0.042 \pm 0.002}$ | $\mathbf{0.97 \pm 0.04}$ | $\mathbf{0.0038 \pm 0.0001}$ |
| Timestepping metric | NMSE ($\downarrow$) | NMSE ($\downarrow$) | NMSE ($\downarrow$) | MSE $\times 10^{-3}$($\downarrow$) |
| Baseline | $0.508 \pm 0.102$ | $0.549 \pm 0.095$ | $0.110 \pm 0.008$ | $2.37 \pm 0.01$ |
| + SSL repr. conditioning | $\mathbf{0.330 \pm 0.081}$ | $\mathbf{0.381 \pm 0.097}$ | $0.108 \pm 0.011$ | $\mathbf{2.35 \pm 0.03}$ |

dataset of only 2000 samples. We compare to the same supervised model (encoder and linear head) trained on the same labeled dataset. The viscosities used range between 0.001 and 0.007 and are sampled uniformly. We can see in Table 1 that we are able to improve over the supervised baseline by leveraging our learned representations. This remains true even when also using Lie Point symmetries for the supervised baselines or when using comparable dataset sizes, as in Figure 4. We also clearly see the ability of our self-supervised approach to leverage larger dataset sizes, whereas we did not see any gain when going to bigger datasets in the supervised setting.

**Initial condition regression (inverse problem):** For the KS and KdV PDEs, we aim to solve the inverse problem by regressing initial condition parameters from a snapshot of future time evolutions of the solution. Following [34, 12], for a domain $\Omega = [0, L]$, a truncated Fourier series, parameterized by $A_k, \omega_k, \phi_k$, is used to generate initial conditions:

$$u_0(x) = \sum_{k=1}^{N} A_k \sin\left(\frac{2\pi\omega_k x}{L} + \phi_k\right). \tag{9}$$

Our task is to regress the set of $2N$ coefficients $\{A_k, \omega_k : k \in \{1, \ldots, N\}\}$ from a snapshot of the solution starting at $t = 20$ to $t = T$. This way, the initial conditions and first-time steps are never seen during training, making the problem non-trivial. For all conducted tests, $N = 10$, $A_k \sim \mathcal{U}(-0.5, 0.5)$, and $\omega_k \sim \mathcal{U}(-0.4, 0.4)$. By neglecting phase shifts, $\phi_k$, the inverse problem is invariant to Galilean boosts and spatial translations, which we use as augmentations for training our SSL method (see Appendix E). The datasets used for KdV and KS contains 10,000 training samples and 2,500 test samples. As shown in Table 1, the SSL trained network reduces NMSE by a factor of almost three compared to the supervised baseline. This demonstrates how pre-training via SSL can help to extract the underlying dynamics from a snapshot of a solution.

**Buoyancy magnitude regression:** Following [18], our dataset consists of solutions of Navier Stokes (Equation (8)) where the external buoyancy force, $\boldsymbol{f} = (c_x, c_y)^\top$, is constant in the two spatial directions over the course of a given evolution, and our aim is to regress the magnitude of this force $\sqrt{c_x^2 + c_y^2}$ given a solution to the PDE. We reuse the dataset generated in [18], where $c_x = 0$ and $c_y \sim \mathcal{U}(0.2, 0.5)$. In practice this gives us 26,624 training samples that we used as our "unlabeled" dataset, 3,328 to train the downstream task on, and 6,592 to evaluate the models. As observed in Table 1, the self-supervised approach is able to significantly outperform the supervised baseline. Even when looking at the best supervised performance (over 60 runs), or in similar data regimes as the supervised baseline illustrated in Fig. 4, the self-supervised baseline consistently performs better and improves further when given larger unlabeled datasets.

Table 2: One-step validation MSE (rescaled by $1e3$) for time-stepping on Navier-Stokes with varying buoyancies for different combinations of architectures and conditioning methods. Architectures are taken from [18] with the same choice of hyper-parameters. Results with ground truth buoyancies are an upper-bound on the performance a representation containing information on the buoyancy.

| Architecture | $UNet_{mod64}$ | $UNet_{mod64}$ | $FNO_{128modes16}$ | $UF1Net_{modes16}$ |
|---|---|---|---|---|
| Conditioning method | Addition [18] | AdaGN [35] | Spatial-Spectral [18] | Addition [18] |
| Time conditioning only | $2.60 \pm 0.05$ | $2.37 \pm 0.01$ | $13.4 \pm 0.5$ | $3.31 \pm 0.06$ |
| Time + SSL repr. cond. | $\mathbf{2.47 \pm 0.02}$ | $\mathbf{2.35 \pm 0.03}$ | $13.0 \pm 1.0$ | $\mathbf{2.37 \pm 0.05}$ |
| Time + true buoyancy cond. | $2.08 \pm 0.02$ | $2.01 \pm 0.02$ | $11.4 \pm 0.8$ | $2.87 \pm 0.03$ |

## 4.2 Time-stepping

To explore whether learned representations improve time-stepping, we study neural networks that use a sequence of time steps (the "history") of a PDE to predict a future sequence of steps. For each equation we consider different conditioning schemes, to fit within the data modality and be comparable to previous work.

**Burgers, Korteweg-de Vries, and Kuramoto-Sivashinsky:** We time-step on 2000 unseen samples for each PDE. To do so, we compute a representation of 20 first input time steps using our frozen encoder, and add it as a new channel. The resulting input is fed to a CNN as in [12] to predict the next 20 time steps (illustrated in Fig. 4 (bottom right) in the context of Burgers' equation). As shown in Table 1, conditioning the neural network or operator with pre-trained representations slightly reduces the error. Such conditioning noticeably improves performance for KdV and KS, while the results are mixed for Burgers'. A potential explanation is that KdV and KS feature more chaotic behavior than Burgers, leaving room for improvement.

**Navier-Stokes' equation:** As pointed out in [18], conditioning a neural network or neural operator on the buoyancy helps generalization accross different values of this parameter. This is done by embedding the buoyancy before mixing the resulting vector either via addition to the neural operator's hidden activations (denoted in [18] as "Addition"), or alternatively for UNets by affine transformation of group normalization layers (denoted as "AdaGN" and originally proposed in [35]). For our main experiment, we use the same modified UNet with 64 channels as in [18] for our neural operator, since it yields the best performance on the Navier-Stokes dataset. To condition the UNet, we compute our representation on the 16 first frames (that are therefore excluded from the training), and pass the representation through a two layer MLP with a bottleneck of size 1, in order to exploit the ability of our representation to recover the buoyancy with only one linear layer. The resulting output is then added to the conditioning embedding as in [18]. Finally, we choose AdaGN as our conditioning method, since it provides the best results in [18]. We follow a similar training and evaluation protocol to [18], except that we perform 20 epochs with cosine annealing schedule on 1,664 trajectories instead of 50 epochs, as we did not observe significant difference in terms of results, and this allowed to explore other architectures and conditioning methods. Additional details are provided in Appendix F. As a baseline, we use the same model without buoyancy conditioning. Both models are conditioned on time. We report the one-step validation MSE on the same time horizons as [18]. Conditioning on our representation outperforms the baseline without conditioning.

We also report results for different architectures and conditioning methods for Navier-Stokes in Table 2 and Burgers in Table 8 (Appendix F.1) validating the potential of conditioning on SSL representations for different models. FNO [36] does not perform as well as other models, partly due to the relatively low number of samples used and the low-resolution nature of the benchmarks. For Navier-Stokes, we also report results obtained when conditioning on both time and ground truth buoyancy, which serves as an upper-bound on the performance of our method. We conjecture these results can be improved by further increasing the quality of the learned representation, *e.g* by training on more samples or through further augmentation tuning. Indeed, the MSE on buoyancy regression obtained by SSL features, albeit significantly lower than the supervised baseline, is often still too imprecise to distinguish consecutive buoyancy values in our data.

## 4.3 Analysis

**Self-supervised learning outperforms supervised learning for PDEs:** While the superiority of self-supervised over supervised representation learning is still an open question in computer vision [37, 38],

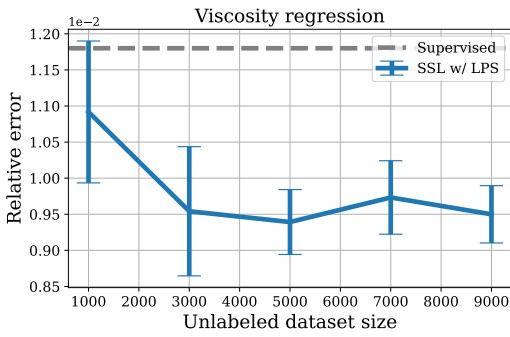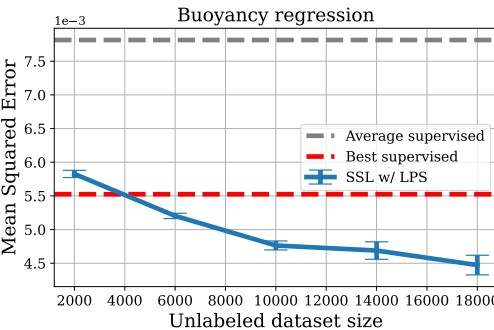

Figure 4: Influence of dataset size on regression tasks. **(Left)** Kinematic regression on Burger's equation. When using Lie point symmetries (LPS) during pretraining, we are able to improve performance over the supervised baselines, even when using an unlabled dataset size that is half the size of the labeled one. As we increase the amount of unlabeled data that we use, the performance improves, further reinforcing the usefulness of self-supervised representations. **(Right)** Buoyancy regression on Navier-Stokes' equation. We notice a similar trend as in Burgers but found that the supervised approach was less stable than the self-supervised one. As such, SSL brings better performance as well as more stability here.

the former outperforms the latter in the PDE domain we consider. A possible explanation is that enforcing similar representations for two different views of the same solution forces the network to learn the underlying dynamics, while the supervised objectives (such as regressing the buoyancy) may not be as informative of a signal to the network. Moreover, Fig. 4 illustrates how more pretraining data benefits our SSL setup, whereas in our experiments it did not help the supervised baselines.

**Cropping:** Cropping is a natural, effective, and popular augmentation in computer vision [21, 39, 40]. In the context of PDE samples, unless specified otherwise, we crop both in temporal and spatial domains finding such a procedure is necessary for the encoder to learn from the PDE data. Cropping also offers a typically weaker means of enforcing analogous space and time translation invariance. The exact size of the crops is generally domain dependent and requires tuning. We quantify its effect in Fig. 5 in the context of Navier-Stokes; here, crops must contain as much information as possible while making sure that pairs of crops have as little overlap as possible (to discourage the network from relying on spurious correlations). This explains the two modes appearing in Fig. 5. We make a similar observation for Burgers, while KdV and KS are less sensitive. Finally, crops help bias the network to learn features that are invariant to whether the input was taken near a boundary or not, thus alleviating the issue of boundary condition preservation during augmentations.

**Selecting Lie point augmentations:** Whereas cropping alone yields satisfactory representations, Lie point augmentations can enhance performance but require careful tuning. In order to choose which symmetries to include in our SSL pipeline and at what strengths to apply them, we study the effectiveness of each Lie augmentation separately. More precisely, given an equation and each possible Lie point augmentation, we train a SSL representation using this augmentation only and cropping. Then, we couple all Lie augmentations improving the representation over simply using crops. In order for this composition to stay in the stability/convergence radius of the Lie Symmetries, we reduce each augmentation's optimal strength by an order of magnitude. Fig. 5 illustrates this process in the context of Navier-Stokes.

# 5 Discussion

This work leverages Lie point symmetries for self-supervised representation learning from PDE data. Our preliminary experiments with the Burgers', KdV, KS, and Navier-Stokes equations demonstrate the usefulness of the resulting representation for sample or compute efficient estimation of characteristics and time-stepping. Nevertheless, a number of limitations are present in this work, which we hope can be addressed in the future. The methodology and experiments in this study were confined to a particular set of PDEs, but we believe they can be expanded beyond our setting.

| Augmentation | Best strength | Buoyancy MSE |
|---|---|---|
| Crop | N.A | $0.0051 \pm 0.0001$ |
| *single Lie transform* | | |
| + $t$ translate $g_1$ | 0.1 | $0.0052 \pm 0.0001$ |
| + $x$ translate $g_2$ | 10.0 | $0.0041 \pm 0.0002$ |
| + scaling $g_4$ | 1.0 | $0.0050 \pm 0.0003$ |
| + rotation $g_5$ | 1.0 | $0.0049 \pm 0.0001$ |
| + boost $g_6$ * | 0.1 | $0.0047 \pm 0.0002$ |
| + boost $g_8$ ** | 0.1 | $0.0046 \pm 0.0001$ |
| *combined* | | |
| + $\{g_2, g_5, g_6, g_8\}$ | *best* / 10 | $0.0038 \pm 0.0001$ |

* linear boost applied in $x$ direction (see Table 7)

** quadratic boost applied in $x$ direction (see Table 7)

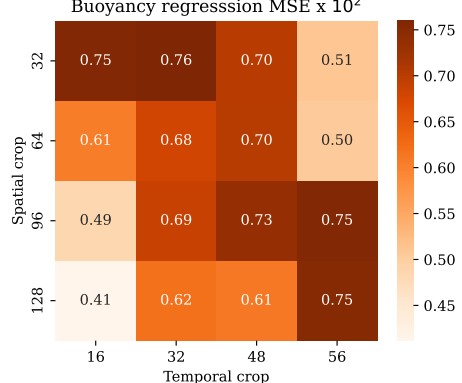

Figure 5: **(Left)** Isolating effective augmentations for Navier-Stokes. Note that we do not study $g_3$, $g_7$ and $g_9$, which are respectively counterparts of $g_2$, $g_6$ and $g_8$ applied in $y$ instead of $x$. **(Right)** Influence of the crop size on performance. We see that performance is maximized when the crops are as large as possible with as little overlap as possible when generating pairs of them.

**Learning equivariant representations:** Another interesting direction is to expand our SSL framework to learning explicitly equivariant features [41, 42]. Learning *equivariant* representations with SSL could be helpful for time-stepping, perhaps directly in the learned representation space.

**Preserving boundary conditions and leveraging other symmetries:** Theoretical insights can also help improve the results contained here. Symmetries are generally derived with respect to systems with infinite domain or periodic boundaries. Since boundary conditions violate such symmetries, we observed in our work that we are only able to implement group operations with small strengths. Finding ways to preserve boundary conditions during augmentation, even approximately, would help expand the scope of symmetries available for learning tasks. Moreover, the available symmetry group operations of a given PDE are not solely comprised of Lie point symmetries. Other types of symmetries, such as nonlocal symmetries or approximate symmetries like Lie-Backlund symmetries, may also be implemented as potential augmentations [13].

**Towards foundation models for PDEs:** A natural next step for our framework is to train a common representation on a mixture of data from different PDEs, such as Burgers, KdV and KS, that are all models of chaotic flow sharing many Lie point symmetries. Our preliminary experiments are encouraging yet suggest that work beyond the scope of this paper is needed to deal with the different time and length scales between PDEs.

**Extension to other scientific data:** In our study, utilizing the structure of PDE solutions as "exact" SSL augmentations for representation learning has shown significant efficacy over supervised methods. This approach's potential extends beyond the PDEs we study as many problems in mathematics, physics, and chemistry present inherent symmetries that can be harnessed for SSL. Future directions could include implementations of SSL for learning stochastic PDEs, or Hamiltonian systems. In the latter, the rich study of Noether's symmetries in relation to Poisson brackets could be a useful setting to study [11]. Real-world data, as opposed to simulated data, may offer a nice application to the SSL setting we study. Here, the exact form of the equation may not be known and symmetries of the equations would have to be garnered from basic physical principles (e.g., flow equations have translational symmetries), derived from conservation laws, or potentially learned from data.

## Acknowledgements

The authors thank Aaron Lou, Johannes Brandstetter, and Daniel Worrall for helpful feedback and discussions. HL is supported by the Fannie and John Hertz Foundation and the NSF Graduate Fellowship under Grant No. 1745302.

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

# Appendix

## Table of Contents

## A  PDE Symmetry Groups and Deriving Generators

Symmetry augmentations encourage invariance of the representations to known symmetry groups of the data. The guiding principle is that inputs that can be obtained from one another via transformations of the symmetry group should share a common representation. In images, such symmetries are known *a priori* and correspond to flips, resizing, or rotations of the input. In PDEs, these symmetry groups can be derived as Lie groups, commonly denoted as Lie point symmetries, and have been categorized for many common PDEs [11]. An example of the form of such augmentations is given in Figure 6 for a simple PDE that rotates a point in 2-D space. In this example, the PDE exhibits both rotational symmetry and scaling symmetry of the radius of rotation. For arbitrary PDEs, such symmetries can be derived, as explained in more detail below.

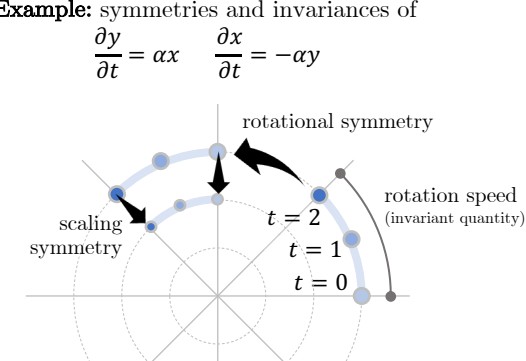

Figure 6: Illustration of the PDE symmetry group and invariances of a simple PDE, which rotates a point in 2-D space. The PDE symmetry group here corresponds to scalings of the radius of the rotation and fixed rotations of all the points over time. A sample invariant quantity is the rate of rotation (related to the parameter $\alpha$ in the PDE), which is fixed for any solution to this PDE.

The Lie point symmetry groups of differential equations form a Lie group structure, where elements of the groups are smooth and differentiable transformations. It is typically easier to derive the symmetries of a system of differential equations via the infinitesimal generators of the symmetries, (*i.e.,* at the level of the derivatives of the one parameter transforms). By using these infinitesimal generators, one can replace *nonlinear* conditions for the invariance of a function under the group transformation, with an equivalent *linear* condition of *infinitesimal* invariance under the respective generator of the group action [11].

In what follows, we give an informal overview to the derivation of Lie point symmetries. Full details and formal rigor can be obtained in Olver [11], Ibragimov [13], among others.

In the setting we consider, a differential equation has a set of $p$ independent variables $\boldsymbol{x} = (x^1, x^2, \ldots, x^p) \in \mathbb{R}^p$ and $q$ dependent variables $\boldsymbol{u} = (u^1, u^2, \ldots, u^q) \in \mathbb{R}^q$. The solutions take the form $\boldsymbol{u} = f(\boldsymbol{x})$, where $u^\alpha = f^\alpha(\boldsymbol{x})$ for $\alpha \in \{1, \ldots, q\}$. Solutions form a graph over a domain $\Omega \subset \mathbb{R}^p$:

$$\Gamma_f = \{(\boldsymbol{x}, f(\boldsymbol{x})) : \boldsymbol{x} \in \Omega\} \subset \mathbb{R}^p \times \mathbb{R}^q. \tag{10}$$

In other words, a given solution $\Gamma_f$ forms a $p$-dimensional submanifold of the space $\mathbb{R}^p \times \mathbb{R}^q$.

The $n$-th **prolongation** of a given smooth function $\Gamma_f$ expands or "prolongs" the graph of the solution into a larger space to include derivatives up to the $n$-th order. More precisely, if $\mathcal{U} = \mathbb{R}^q$ is the solution space of a given function and $f : \mathbb{R}^p \to \mathcal{U}$, then we introduce the Cartesian product space of the prolongation:

$$\mathcal{U}^{(n)} = \mathcal{U} \times \mathcal{U}_1 \times \mathcal{U}_2 \times \cdots \times \mathcal{U}_n, \tag{11}$$

where $\mathcal{U}_k = \mathbb{R}^{\dim(k)}$ and $\dim(k) = \binom{p+k-1}{k}$ is the dimension of the so-called *jet space* consisting of all $k$-th order derivatives. Given any solution $f : \mathbb{R}^p \to \mathcal{U}$, the prolongation can be calculated by simply calculating the corresponding derivatives up to order $n$ (e.g., via a Taylor expansion at each point). For a given function $\boldsymbol{u} = f(\boldsymbol{x})$, the $n$-th prolongation is denoted as $\boldsymbol{u}^{(n)} = \mathrm{pr}^{(n)} f(\boldsymbol{x})$. As a simple example, for the case of $p = 2$ with independent variables $x$ and $y$ and $q = 1$ with a single

dependent variable $f$, the second prolongation is

$$
\begin{aligned}
\boldsymbol{u}^{(2)} = \mathrm{pr}^{(2)} f(x,y) &= (u; u_x, u_y; u_{xx}, u_{xy}, u_{yy}) \\
&= \left( f; \frac{\partial f}{\partial x}, \frac{\partial f}{\partial y}; \frac{\partial^2 f}{\partial x^2}, \frac{\partial^2 f}{\partial x \partial y}, \frac{\partial^2 f}{\partial y^2} \right) \in \mathbb{R}^1 \times \mathbb{R}^2 \times \mathbb{R}^3,
\end{aligned}
\tag{12}
$$

which is evaluated at a given point $(x,y)$ in the domain. The complete space $\mathbb{R}^p \times \mathcal{U}^{(n)}$ is often called the $n$-th order jet space [11].

A system of differential equations is a set of $l$ differential equations $\Delta : \mathbb{R}^p \times \mathcal{U}^{(n)} \to \mathbb{R}^l$ of the independent and dependent variables with dependence on the derivatives up to a maximum order of $n$:

$$
\Delta_\nu(\boldsymbol{x}, \boldsymbol{u}^{(n)}) = 0, \quad \nu = 1, \dots, l.
\tag{13}
$$

A smooth solution is thus a function $f$ such that for all points in the domain of $\boldsymbol{x}$:

$$
\Delta_\nu(\boldsymbol{x}, \mathrm{pr}^{(n)} f(\boldsymbol{x})) = 0, \quad \nu = 1, \dots, l.
\tag{14}
$$

In geometric terms, the system of differential equations states where the given map $\Delta$ vanishes on the jet space, and forms a subvariety

$$
Z_\Delta = \{(\boldsymbol{x}, \boldsymbol{u}^{(n)}) : \Delta(\boldsymbol{x}, \boldsymbol{u}^{(n)}) = 0\} \subset \mathbb{R}^p \times \mathcal{U}^{(n)}.
\tag{15}
$$

Therefore to check if a solution is valid, one can check if the prolongation of the solution falls within the subvariety $Z_\Delta$. As an example, consider the one dimensional heat equation

$$
\Delta = u_t - cu_{xx} = 0.
\tag{16}
$$

We can check that $f(x,t) = \sin(x)e^{-ct}$ is a solution by forming its prolongation and checking if it falls withing the subvariety given by the above equation:

$$
\mathrm{pr}^{(2)} f(x,t) = \left( \sin(x)e^{-ct}; \cos(x)e^{-ct}, -c\sin(x)e^{-ct}; -\sin(x)e^{-ct}, -c\cos(x)e^{-ct}, c^2\sin(x)e^{-ct} \right),
$$
$$
\Delta(x,t,\boldsymbol{u}^{(n)}) = -c\sin(x)e^{-ct} + c\sin(x)e^{-ct} = 0.
\tag{17}
$$

## A.1 Symmetry Groups and Infinitesimal Invariance

A symmetry group $G$ for a system of differential equations is a set of local transformations to the function which transform one solution of the system of differential equations to another. The group takes the form of a Lie group, where group operations can be expressed as a composition of one-parameter transforms. More rigorously, given the graph of a solution $\Gamma_f$ as defined in Eq. (10), a group operation $g \in G$ maps this graph to a new graph

$$
g \cdot \Gamma_f = \{(\tilde{\boldsymbol{x}}, \tilde{\boldsymbol{u}}) = g \cdot (\boldsymbol{x}, \boldsymbol{u}) : (\boldsymbol{x}, \boldsymbol{u}) \in \Gamma_f\},
\tag{18}
$$

where $(\tilde{\boldsymbol{x}}, \tilde{\boldsymbol{u}})$ label the new coordinates of the solution in the set $g \cdot \Gamma_f$. For example, if $\boldsymbol{x} = (x,t)$, $u = u(x,t)$, and $g$ acts on $(\boldsymbol{x}, u)$ via

$$
(x, t, u) \mapsto (x + \epsilon t, t, u + \epsilon),
$$

then $\tilde{u}(\tilde{x}, \tilde{t}) = u(x,t) + \epsilon = u(\tilde{x} - \epsilon\tilde{t}, \tilde{t}) + \epsilon$, where $(\tilde{x}, \tilde{t}) = (x + \epsilon t, t)$.

Note, that the set $g \cdot \Gamma_f$ may not necessarily be a graph of a new $\boldsymbol{x}$-valued function; however, since all transformations are local and smooth, one can ensure transformations are valid in some region near the identity of the group.

As an example, consider the following transformations which are members of the symmetry group of the differential equation $u_{xx} = 0$. $g_1(t)$ translates a single spatial coordinate $x$ by an amount $t$ and $g_2$ scales the output coordinate $u$ by an amount $e^r$:

$$
\begin{aligned}
g_1(t) \cdot (x, u) &= (x + t, u), \\
g_2(r) \cdot (x, u) &= (x, e^r \cdot u).
\end{aligned}
\tag{19}
$$

It is easy to verify that both of these operations are local and smooth around a region of the identity, as sending $r, t \to 0$ recovers the identity operation. Lie theory allows one to equivalently describe

the potentially nonlinear group operations above with corresponding infinitesimal generators of the group action, corresponding to the Lie algebra of the group. Infinitesimal generators form a vector field over the total space $\Omega \times \mathcal{U}$, and the group operations correspond to integral flows over that vector field. To map from a single parameter Lie group operation to its corresponding infinitesimal generator, we take the derivative of the single parameter operation at the identity:

$$\boldsymbol{v}_g|_{(x,u)} = \frac{d}{dt} g(t) \cdot (x, u)\Big|_{t=0}, \tag{20}$$

where $g(0) \cdot (x, u) = (x, u)$.

To map from the infinitesimal generator back to the corresponding group operation, one can apply the exponential map

$$\exp(t\boldsymbol{v}) \cdot (\boldsymbol{x}, \boldsymbol{u}) = g(t) \cdot (\boldsymbol{x}, \boldsymbol{u}), \tag{21}$$

where $\exp : \mathfrak{g} \to G$. Here, $\exp(\cdot)$ maps from the Lie algebra, $\mathfrak{g}$, to the corresponding Lie group, $G$. This exponential map can be evaluated using various methods, as detailed in Appendix B and Appendix E.

Returning to the example earlier from Equation (19), the corresponding Lie algebra elements are

$$\begin{aligned}
\boldsymbol{v}_{g_1} &= \partial_x \leftrightarrow g_1(t) \cdot (x, u) = (x + t, u), \\
\boldsymbol{v}_{g_2} &= u\partial_u \leftrightarrow g_2(r) \cdot (x, u) = (x, e^r \cdot u).
\end{aligned} \tag{22}$$

Informally, Lie algebras help simplify notions of invariance as it allows one to check whether functions or differential equations are invariant to a group by needing only to check it at the level of the derivative of that group. In other words, for any vector field corresponding to a Lie algebra element, a given function is invariant to that vector field if the action of the vector field on the given function evaluates to zero everywhere. Thus, given a symmetry group, one can determine a set of invariants using the vector fields corresponding to the infinitesimal generators of the group. To determine whether a differential equation is in such a set of invariants, we extend the definition of a prolongation to act on vector fields as

$$\mathrm{pr}^{(n)}\, \boldsymbol{v}\big|_{(\boldsymbol{x},\boldsymbol{u}^{(n)})} = \frac{d}{d\epsilon}\Big|_{\epsilon=0} \mathrm{pr}^{(n)}\left[\exp(\epsilon\boldsymbol{v})\right](\boldsymbol{x}, \boldsymbol{u}^{(n)}). \tag{23}$$

A given vector field $\boldsymbol{v}$ is therefore an infinitesimal generator of a symmetry group $G$ of a system of differential equations $\Delta_\nu$ indexed by $\nu \in \{1, \ldots, l\}$ if the prolonged vector field of any given solution is still a solution:

$$\mathrm{pr}^{(n)}\, \boldsymbol{v}[\Delta_\nu(\boldsymbol{x}, \boldsymbol{u}^{(n)})] = 0, \quad \nu = 1, \ldots, l, \quad \text{whenever } \Delta(\boldsymbol{x}, \boldsymbol{u}^{(n)}) = 0. \tag{24}$$

For sake of convenience and brevity, we leave out many of the formal definitions behind these concepts and refer the reader to [11] for complete details.

## A.2 Deriving Generators of the Symmetry Group of a PDE

Since symmetries of differential equations correspond to smooth maps, it is typically easier to derive the particular symmetries of a differential equation via their infinitesimal generators. To derive such generators, we first show how to perform the prolongation of a vector field. As before, assume we have $p$ independent variables $x^1, \ldots, x^p$ and $l$ dependent variables $u^1, \ldots, u^l$, which are a function of the dependent variables. Note that we use superscripts to denote a particular variable. Derivatives with respect to a given variable are denoted via subscripts corresponding to the indices. For example, the variable $u^1_{112}$ denotes the third order derivative of $u^1$ taken twice with respect to the variable $x^1$ and once with respect to $x^2$. As stated earlier, the prolongation of a vector field is defined as the operation

$$\mathrm{pr}^{(n)}\, \boldsymbol{v}\big|_{(\boldsymbol{x},\boldsymbol{u}^{(n)})} = \frac{d}{d\epsilon}\Big|_{\epsilon=0} \mathrm{pr}^{(n)}\left[\exp(\epsilon\boldsymbol{v})\right](\boldsymbol{x}, \boldsymbol{u}^{(n)}). \tag{25}$$

To calculate the above, we can evaluate the formula on a vector field written in a generalized form. *I.e.*, any vector field corresponding to the infinitesimal generator of a symmetry takes the general form

$$\boldsymbol{v} = \sum_{i=1}^p \xi^i(\boldsymbol{x}, \boldsymbol{u}) \frac{\partial}{\partial x^i} + \sum_{\alpha=1}^q \phi_\alpha(\boldsymbol{x}, \boldsymbol{u}) \frac{\partial}{\partial u^\alpha}. \tag{26}$$

Throughout, we will use Greek letter indices for dependent variables and standard letter indices for independent variables. Then, we have that

$$\mathrm{pr}^{(n)}\,\boldsymbol{v} = \boldsymbol{v} + \sum_{\alpha=1}^{q}\sum_{\boldsymbol{J}} \phi_{\alpha}^{\boldsymbol{J}}(\boldsymbol{x}, \boldsymbol{u}^{(n)})\frac{\partial}{\partial u_{\boldsymbol{J}}^{\alpha}}, \tag{27}$$

where $\boldsymbol{J}$ is a tuple of dependent variables indicating which variables are in the derivative of $\frac{\partial}{\partial u_{\boldsymbol{J}}^{\alpha}}$. Each $\phi_{\alpha}^{\boldsymbol{J}}(\boldsymbol{x}, \boldsymbol{u}^{(n)})$ is calculated as

$$\phi_{\alpha}^{\boldsymbol{J}}(\boldsymbol{x}, \boldsymbol{u}^{(n)}) = \prod_{i \in \boldsymbol{J}} \boldsymbol{D}_i\left(\phi_a - \sum_{i=1}^{p}\xi^i u_i^{\alpha}\right) + \sum_{i=1}^{p}\xi^i u_{\boldsymbol{J},i}^{\alpha}, \tag{28}$$

where $u_{\boldsymbol{J},i}^{\alpha} = \partial u_{\boldsymbol{J}}^{\alpha}/\partial x^i$ and $\boldsymbol{D}_i$ is the total derivative operator with respect to variable $i$ defined as

$$\boldsymbol{D}_i P(x, u^{(n)}) = \frac{\partial P}{\partial x^i} + \sum_{i=1}^{q}\sum_{\boldsymbol{J}} u_{\boldsymbol{J},i}^{\alpha}\frac{\partial P}{\partial u_{\boldsymbol{J}}^{\alpha}}. \tag{29}$$

After evaluating the coefficients, $\phi_{\alpha}^{\boldsymbol{J}}(x, u^{(n)})$, we can substitute these values into the definition of the vector field's prolongation in Equation (27). This fully describes the infinitesimal generator of the given PDE, which can be used to evaluate the necessary symmetries of the system of differential equations. An example for Burgers' equation, a canonical PDE, is presented in the following.

### A.3 Example: Burgers' Equation

Burgers' equation is a PDE used to describe convection-diffusion phenomena commonly observed in fluid mechanics, traffic flow, and acoustics [43]. The PDE can be written in either its "potential" form or its "viscous" form. The potential form is

$$u_t = u_{xx} + u_x^2. \tag{30}$$

**Cautionary note:** We derive here the symmetries of Burgers' equation in its potential form since this form is more convenient and simpler to study for the sake of an example. The equation we consider in our experiments is the more commonly studied Burgers' equation in its standard form which does not have the same Lie symmetry group (see Table 4). Similar derivations for Burgers' equation in its standard form can be found in example 6.1 of [44].

Following the notation from the previous section, $p = 2$ and $q = 1$. Consequently, the symmetry group of Burgers' equation will be generated by vector fields of the following form

$$\boldsymbol{v} = \xi(x, t, u)\frac{\partial}{\partial x} + \tau(x, t, u)\frac{\partial}{\partial t} + \phi(x, t, u)\frac{\partial}{\partial u}, \tag{31}$$

where we wish to determine all possible coefficient functions, $\xi(t, x, u)$, $\tau(x, t, u)$, and $\phi(x, t, u)$ such that the resulting one-parameter sub-group $\exp(\varepsilon\mathbf{v})$ is a symmetry group of Burgers' equation.

To evaluate these coefficients, we need to prolong the vector field up to 2$^{\mathrm{nd}}$ order, given that the highest-degree derivative present in the governing PDE is of order 2. The 2$^{\mathrm{nd}}$ prolongation of the vector field can be expressed as

$$\mathrm{pr}^{(2)}\,\boldsymbol{v} = \boldsymbol{v} + \phi^x\frac{\partial}{\partial u_x} + \phi^t\frac{\partial}{\partial u_t} + \phi^{xx}\frac{\partial}{\partial u_{xx}} + \phi^{xt}\frac{\partial}{\partial u_{xt}} + \phi^{tt}\frac{\partial}{\partial u_{tt}}. \tag{32}$$

Applying this prolonged vector field to the differential equation in Equation (30), we get the infinitesimal symmetry criteria that

$$\mathrm{pr}^{(2)}\,\boldsymbol{v}[\Delta(x, t, \boldsymbol{u}^{(2)})] = \phi^t - \phi^{xx} + 2u_x\phi^x = 0. \tag{33}$$

To evaluate the individual coefficients, we apply Equation (28). Next, we substitute every instance of $u_t$ with $u_x^2 + u_{xx}$, and equate the coefficients of each monomial in the first and second-order

Table 3: Monomial coefficients in vector field prolongation for Burgers' equation.

| Monomial | Coefficient |
|---|---|
| $1$ | $\phi_t = \phi_{xx}$ |
| $u_x$ | $2\phi_x + 2(\phi_{xu} - \xi_{xx}) = -\xi_t$ |
| $u_x^2$ | $2(\phi_u - \xi_x) - \tau_{xx} + (\phi_{uu} - 2\xi_{xu}) = \phi_u - \tau_t$ |
| $u_x^3$ | $-2\tau_x - 2\xi_u - 2\tau_{xu} - \xi_{uu} = -\xi_u$ |
| $u_x^4$ | $-2\tau_u - \tau_{uu} = -\tau_u$ |
| $u_{xx}$ | $-\tau_{xx} + (\phi_u - 2\xi_x) = \phi_u - \tau_t$ |
| $u_x u_{xx}$ | $-2\tau_x - 2\tau_{xu} - 3\xi_u = -\xi_u$ |
| $u_x^2 u_{xx}$ | $-2\tau_u - \tau_{uu} - \tau_u = -2\tau_u$ |
| $u_{xx}^2$ | $-\tau_u = -\tau_u$ |
| $u_{xt}$ | $-2\tau_x = 0$ |
| $u_x u_{xt}$ | $-2\tau_u = 0$ |

derivatives of $u$ to find the pertinent symmetry groups. Table 3 below lists the relevant monomials as well as their respective coefficients.

Using these relations, we can solve for the coefficient functions. For the case of Burgers' equation, the most general infinitesimal symmetries have coefficient functions of the following form:

$$\xi(t, x) = k_1 + k_4 x + 2k_5 t + 4k_6 xt \tag{34}$$

$$\tau(t) = k_2 + 2k_4 t + 4k_6 t^2 \tag{35}$$

$$\phi(t, x, u) = (k_3 - k_5 x - 2k_6 t - k_6 x^2)u + \gamma(x, t) \tag{36}$$

where $k_1, \ldots, k_6 \in \mathbb{R}$ and $\gamma(x, t)$ is an arbitrary solution to Burgers' equation. These coefficient functions can be used to generate the infinitesimal symmetries. These symmetries are spanned by the six vector fields below:

$$\boldsymbol{v}_1 = \partial_x \tag{37}$$

$$\boldsymbol{v}_2 = \partial_t \tag{38}$$

$$\boldsymbol{v}_3 = \partial_u \tag{39}$$

$$\boldsymbol{v}_4 = x\partial_x + 2t\partial_t \tag{40}$$

$$\boldsymbol{v}_5 = 2t\partial_x - x\partial_u \tag{41}$$

$$\boldsymbol{v}_6 = 4xt\partial_x + 4t^2\partial_t - (x^2 + 2t)\partial_u \tag{42}$$

as well as the infinite-dimensional subalgebra: $\boldsymbol{v}_\gamma = \gamma(x, t)e^{-u}\partial_u$. Here, $\gamma(x, t)$ is any arbitrary solution to the heat equation. The relationship between the Heat equation and Burgers' equation can be seen, whereby if $u$ is replaced by $w = e^u$, the Cole–Hopf transformation is recovered.

## B  Exponential map and its approximations

As observed in the previous section, symmetry groups are generally derived in the Lie algebra of the group. The exponential map can then be applied, taking elements of this Lie algebra to the corresponding group operations. Working within the Lie algebra of a group provides several benefits. First, a Lie algebra is a vector space, so elements of the Lie algebra can be added and subtracted to yield new elements of the Lie algebra (and the group, via the exponential map). Second, when generators of the Lie algebra are closed under the Lie bracket of the Lie algebra (*i.e.*, the generators form a basis for the structure constants of the Lie algebra), any arbitrary Lie point symmetry can be obtained via an element of the Lie algebra (i.e. the exponential map is surjective onto the connected component of the identity) [11]. In contrast, composing group operations in an arbitrary, *fixed* sequence is not guaranteed to be able to generate any element of the group. Lastly, although not extensively detailed here, the "strength," or magnitude, of Lie algebra elements can be measured using an appropriately selected norm. For instance, the operator norm of a matrix could be used for matrix Lie algebras.

In certain cases, especially when the element $\boldsymbol{v}$ in the Lie algebra consists of a single basis element, the exponential map $\exp(\boldsymbol{v})$ applied to that element of the Lie algebra can be calculated explicitly. Here, applying the group operation to a tuple of independent and dependent variables results in the so-called Lie point transformation, since it is applied at a given point $\exp(\epsilon\boldsymbol{v}) \cdot (x, f(x)) \mapsto (x', f(x)')$. Consider the concrete example below from Burger's equation.

**Example B.1** (Exponential map on symmetry generator of Burger's equation). *The Burger's equation contains the Lie point symmetry $\boldsymbol{v}_\gamma = \gamma(x,t)e^{-u}\partial_u$ with corresponding group transformation $\exp(\epsilon\boldsymbol{v}_\gamma) \cdot (x,t,u) = (x,t,\log(e^u + \epsilon\gamma))$.*

*Proof.* This transformation only changes the $u$ component. Here, we have

$$\exp\left(\epsilon\gamma e^{-u}\partial_u\right)u = u + \sum_{k=1}^{n}\left(\epsilon\gamma e^{-u}\partial_u\right)^k \cdot u \tag{43}$$

$$= u + \epsilon\gamma e^{-u} - \frac{1}{2}\epsilon^2\gamma^2 e^{-2u} + \frac{1}{3}\epsilon^3\gamma^3 e^{-3u} + \cdots$$

Applying the series expansion $\log(1+x) = x - \frac{x^2}{2} + \frac{x^3}{3} - \cdots$, we get

$$\exp\left(\epsilon\gamma e^{-u}\partial_u\right)u = u + \log\left(1 + \epsilon\gamma e^{-u}\right)$$

$$= \log(e^u) + \log\left(1 + \epsilon\gamma e^{-u}\right) \tag{44}$$

$$= \log(e^u + \epsilon\gamma).$$

$\square$

In general, the output of the exponential map cannot be easily calculated as we did above, especially if the vector field $\boldsymbol{v}$ is a weighted sum of various generators. In these cases, we can still apply the exponential map to a desired accuracy using efficient approximation methods, which we discuss next.

### B.1 Approximations to the exponential map

For arbitrary Lie groups, computing the exact exponential map is often not feasible due to the complex nature of the group and its associated Lie algebra. Hence, it is necessary to approximate the exponential map to obtain useful results. Two common methods for approximating the exponential map are the truncation of Taylor series and Lie-Trotter approximations.

**Taylor series approximation** Given a vector field $\boldsymbol{v}$ in the Lie algebra of the group, the exponential map can be approximated by truncating the Taylor series expansion of $\exp(\boldsymbol{v})$. The Taylor series expansion of the exponential map is given by:

$$\exp(\boldsymbol{v}) = \text{Id} + \boldsymbol{v} + \frac{1}{2}\boldsymbol{v} \cdot \boldsymbol{v} + \cdots = \sum_{n=0}^{\infty}\frac{\boldsymbol{v}^n}{n!}. \tag{45}$$

To approximate the exponential map, we retain a finite number of terms in the series:

$$\exp(\boldsymbol{v}) = \sum_{n=0}^{k}\frac{\boldsymbol{v}^n}{n!} + o(\|\boldsymbol{v}\|^k), \tag{46}$$

where $k$ is the order of the truncation. The accuracy of the approximation depends on the number of terms retained in the truncated series and the operator norm $\|\boldsymbol{v}\|$. For matrix Lie groups, where $\boldsymbol{v}$ is also a matrix, this operator norm is equivalent to the largest magnitude of the eigenvalues of the matrix [45]. The error associated with truncating the Taylor series after $k$ terms thus decays exponentially with the order of the approximation.

Two drawbacks exist when using the Taylor approximation. First, for a given vector field $\boldsymbol{v}$, applying $\boldsymbol{v} \cdot f$ to a given function $f$ requires algebraic computation of derivatives. Alternatively, derivatives can also be approximated through finite difference schemes, but this would add an additional source of error. Second, when using the Taylor series to apply a symmetry transformation of a PDE to a starting solution of that PDE, the Taylor series truncation will result in a new function, which is not necessarily a solution of the PDE anymore (although it can be made arbitrarily close to a solution by increasing the truncation order). Lie-Trotter approximations, which we study next, approximate the exponential map by a composition of symmetry operations, thus avoiding these two drawbacks.

**Lie-Trotter series approximations** The Lie-Trotter approximation is an alternative method for approximating the exponential map, particularly useful when one has access to group elements directly, i.e. the closed-form output of the exponential map on each Lie algebra generator), but they are non-commutative. To provide motivation for this method, consider two elements $\boldsymbol{X}$ and $\boldsymbol{Y}$ in the Lie algebra. The Lie-Trotter formula (or Lie product formula) approximates the exponential of their sum [22, 46].

$$\exp(\boldsymbol{X} + \boldsymbol{Y}) = \lim_{n \to \infty} \left[ \exp\left(\frac{\boldsymbol{X}}{n}\right) \exp\left(\frac{\boldsymbol{Y}}{n}\right) \right]^n \approx \left[ \exp\left(\frac{\boldsymbol{X}}{k}\right) \exp\left(\frac{\boldsymbol{Y}}{k}\right) \right]^k, \quad (47)$$

where $k$ is a positive integer controlling the level of approximation.

The first-order approximation above can be extended to higher orders, referred to as the Lie-Trotter-Suzuki approximations. Though various different such approximations exist, we particularly use the following recursive approximation scheme [47, 23] for a given Lie algebra component $\boldsymbol{v} = \sum_{i=1}^{p} \boldsymbol{v}_i$.

$$\mathcal{T}_2(\boldsymbol{v}) = \exp\left(\frac{\boldsymbol{v}_1}{2}\right) \cdot \exp\left(\frac{\boldsymbol{v}_2}{2}\right) \cdots \exp\left(\frac{\boldsymbol{v}_p}{2}\right) \exp\left(\frac{\boldsymbol{v}_p}{2}\right) \cdot \exp\left(\frac{\boldsymbol{v}_{p-1}}{2}\right) \cdots \exp\left(\frac{\boldsymbol{v}_1}{2}\right),$$

$$\mathcal{T}_{2k}(\boldsymbol{v}) = \mathcal{T}_{2k-2}(u_k \boldsymbol{v})^2 \cdot \mathcal{T}_{2k-2}((1 - 4u_k)\boldsymbol{v}) \cdot \mathcal{T}_{2k-2}(u_k \boldsymbol{v})^2, \quad (48)$$

$$u_k = \frac{1}{4 - 4^{1/(2k-1)}}.$$

To apply the above formula, we tune the order parameter $p$ and split the time evolution into $r$ segments to apply the approximation $\exp(\boldsymbol{v}) \approx \prod_{i=1}^{r} \mathcal{T}_p(\boldsymbol{v}/r)$. For the $p$-th order, the number of stages in the Suzuki formula above is equal to $2 \cdot 5^{p/2-1}$, so the total number of stages applied is equal to $2r \cdot 5^{p/2-1}$.

These methods are especially useful in the context of PDEs, as they allow for the approximation of the exponential map while preserving the structure of the Lie algebra and group. Similar techniques are used in the design of splitting methods for numerically solving PDEs [48, 49]. Crucially, these approximations will always provide valid solutions to the PDEs, since each individual group operation in the composition above is itself a symmetry of the PDE. This is in contrast with approximations via Taylor series truncation, which only provide approximate solutions.

As with the Taylor series approximation, the $p$-th order approximation above is accurate to $o(\|\boldsymbol{v}\|^p)$ with suitably selected values of $r$ and $p$ [23]. As a cautionary note, the approximations here may fail to converge when applied to unbounded operators [50, 51]. In practice, we tested a range of bounds to the augmentations and tuned augmentations accordingly (see Appendix E).

## C  VICReg Loss

In our implementations, we use the VICReg loss as our choice of SSL loss [9]. This loss contains three different terms: a variance term that ensures representations do not collapse to a single point, a covariance term that ensures different dimensions of the representation encode different data, and an invariance term to enforce similarity of the representations for pairs of inputs related by an augmentation. We go through each term in more detail below. Given a distribution $\mathcal{T}$ from which to draw augmentations and a set of inputs $\boldsymbol{x}_i$, the precise algorithm to calculate the VICReg loss for a batch of data is also given in Algorithm 1.

Formally, define our embedding matrices as $\boldsymbol{Z}, \boldsymbol{Z}' \in \mathbb{R}^{N \times D}$. Next, we define the similarity criterion, $\mathcal{L}_{\text{sim}}$, as

$$\mathcal{L}_{\text{sim}}(\boldsymbol{u}, \boldsymbol{v}) = \|\boldsymbol{u} - \boldsymbol{v}\|_2^2,$$

which we use to match our embeddings, and to make them invariant to the transformations. To avoid a collapse of the representations, we use the original variance and covariance criteria to define our regularisation loss, $\mathcal{L}_{\text{reg}}$, as

$$\mathcal{L}_{\text{reg}}(\boldsymbol{Z}) = \lambda_{cov}\, C(\boldsymbol{Z}) + \lambda_{var}\, V(\boldsymbol{Z}), \quad \text{with}$$

$$C(\boldsymbol{Z}) = \frac{1}{D} \sum_{i \neq j} \text{Cov}(\boldsymbol{Z})_{i,j}^2 \quad \text{and}$$

$$V(\boldsymbol{Z}) = \frac{1}{D} \sum_{j=1}^{D} \max\left(0, 1 - \sqrt{\text{Var}(\boldsymbol{Z}_{:,j})}\right).$$

**Algorithm 1** VICReg Loss Evaluation

---

**Hyperparameters:** $\lambda_{var}, \lambda_{cov}, \lambda_{inv}, \gamma \in \mathbb{R}$
**Input:** $N$ inputs in a batch $\{\boldsymbol{x}_i \in \mathbb{R}^{D_{in}}, i = 1, \ldots, N\}$
**VICRegLoss($N, \boldsymbol{x}_i, \lambda_{var}, \lambda_{cov}, \lambda_{inv}, \gamma$):**

1: Apply augmentations $t, t' \sim \mathcal{T}$ to form embedding matrices $\boldsymbol{Z}, \boldsymbol{Z}' \in \mathbb{R}^{N \times D}$:

$$\boldsymbol{Z}_{i,:} = h_\theta \left( f_\theta \left( t \cdot \boldsymbol{x}_i \right) \right) \text{ and } \boldsymbol{Z}'_{i,:} = h_\theta \left( f_\theta \left( t' \cdot \boldsymbol{x}_i \right) \right)$$

2: Form covariance matrices $\text{Cov}(\boldsymbol{Z}), \text{Cov}(\boldsymbol{Z}') \in \mathbb{R}^{D \times D}$:

$$\text{Cov}(\boldsymbol{Z}) = \frac{1}{N-1} \sum_{i=1}^{N} \left( \boldsymbol{Z}_{i,:} - \overline{\boldsymbol{Z}}_{i,:} \right) \left( \boldsymbol{Z}_{i,:} - \overline{\boldsymbol{Z}}_{i,:} \right)^\top, \quad \overline{\boldsymbol{Z}}_{i,:} = \frac{1}{N} \sum_{i=1}^{N} \boldsymbol{Z}_{i,:}$$

3: Evaluate loss: $\mathcal{L}(\boldsymbol{Z}, \boldsymbol{Z}') = \lambda_{var} \mathcal{L}_{var}(\boldsymbol{Z}, \boldsymbol{Z}') + \lambda_{cov} \mathcal{L}_{cov}(\boldsymbol{Z}, \boldsymbol{Z}') + \lambda_{inv} \mathcal{L}_{inv}(\boldsymbol{Z}, \boldsymbol{Z}')$

$$\mathcal{L}_{var}(\boldsymbol{Z}, \boldsymbol{Z}') = \frac{1}{D} \sum_{i=1}^{N} \max(0, \gamma - \sqrt{\text{Cov}(\boldsymbol{Z})_{ii}}) + \max(0, \gamma - \sqrt{\text{Cov}(\boldsymbol{Z}')_{ii}}),$$

$$\mathcal{L}_{cov}(\boldsymbol{Z}, \boldsymbol{Z}') = \frac{1}{D} \sum_{i,j=1,i\neq j}^{N} [\text{Cov}(\boldsymbol{Z})_{ij}]^2 + [\text{Cov}(\boldsymbol{Z}')_{ij}]^2,$$

$$\mathcal{L}_{inv}(\boldsymbol{Z}, \boldsymbol{Z}') = \frac{1}{N} \sum_{i=1}^{N} \|\boldsymbol{Z}_{i,:} - \boldsymbol{Z}_{i',:}\|^2$$

4: **Return:** $\mathcal{L}(\boldsymbol{Z}, \boldsymbol{Z}')$

---

The variance criterion, $V(\boldsymbol{Z})$, ensures that all dimensions in the representations are used, while also serving as a normalization of the dimensions. The goal of the covariance criterion is to decorrelate the different dimensions, and thus, spread out information across the embeddings.

The final criterion is

$$\mathcal{L}_{\text{VICReg}}(\boldsymbol{Z}, \boldsymbol{Z}') = \lambda_{\text{inv}} \frac{1}{N} \sum_{i=1}^{N} \mathcal{L}_{\text{sim}}(\boldsymbol{Z}_{i,\text{inv}}, \boldsymbol{Z}'_{i,\text{inv}}) + \mathcal{L}_{\text{reg}}(\boldsymbol{Z}') + \mathcal{L}_{\text{reg}}(\boldsymbol{Z}).$$

Hyperparameters $\lambda_{var}, \lambda_{cov}, \lambda_{inv}, \gamma \in \mathbb{R}$ weight the contributions of different terms in the loss. For all studies conducted in this work, we use the default values of $\lambda_{var} = \lambda_{\text{inv}} = 25$ and $\lambda_{cov} = 1$, unless specified. In our experience, these default settings perform generally well.

## D   Expanded related work

**Machine Learning for PDEs**   Recent work on machine learning for PDEs has considered both invariant prediction tasks [52] and time-series modelling [53, 54]. In the fluid mechanics setting, models learn dynamic viscosities, fluid densities, and/or pressure fields from both simulation and real-world experimental data [55, 56, 57]. For time-dependent PDEs, prior work has investigated the efficacy of convolutional neural networks (CNNs), recurrent neural networks (RNNs), graph neural networks (GNNs), and transformers in learning to evolve the PDE forward in time [34, 58, 59, 60]. This has invoked interest in the development of reduced order models and learned representations for time integration that decrease computational expense, while attempting to maintain solution accuracy. Learning representations of the governing PDE can enable time-stepping in a latent space, where the computational expense is substantially reduced [61]. Recently, for example, Lusch et al. have studied learning the infinite-dimensional Koopman operator to globally linearize latent space dynamics [62]. Kim et al. have employed the Sparse Identification of Nonlinear Dynamics (SINDy) framework to parameterize latent space trajectories and combine them with classical ODE solvers to integrate latent space coordinates to arbitrary points in time [53]. Nguyen et al. have looked at the development of foundation models for climate sciences using transformers pre-trained on well-established climate

datasets [7]. Other methods like dynamic mode decomposition (DMD) are entirely data-driven, and find the best operator to estimate temporal dynamics [63]. Recent extensions of this work have also considered learning equivalent operators, where physical constraints like energy conservation or the periodicity of the boundary conditions are enforced [29].

**Self-supervised learning** All joint embedding self-supervised learning methods have a similar objective: forming representations across a given domain of inputs that are invariant to a certain set of transformations. Contrastive and non-contrastive methods are both used. Contrastive methods [21, 64, 65, 66, 67] push away unrelated pairs of augmented datapoints, and frequently rely on the InfoNCE criterion [68], although in some cases, squared similarities between the embeddings have been employed [69]. Clustering-based methods have also recently emerged [70, 71, 6], where instead of contrasting pairs of samples, samples are contrasted with cluster centroids. Non-contrastive methods [10, 40, 9, 72, 73, 74, 39] aim to bring together embeddings of positive samples. However, the primary difference between contrastive and non-contrastive methods lies in how they prevent representational collapse. In the former, contrasting pairs of examples are explicitly pushed away to avoid collapse. In the latter, the criterion considers the set of embeddings as a whole, encouraging information content maximization to avoid collapse. For example, this can be achieved by regularizing the empirical covariance matrix of the embeddings. While there can be differences in practice, both families have been shown to lead to very similar representations [16, 75]. An intriguing feature in many SSL frameworks is the use of a projector neural network after the encoder, on top of which the SSL loss is applied. The projector was introduced in [21]. Whereas the projector is not necessary for these methods to learn a satisfactory representation, it is responsible for an important performance increase. Its exact role is an object of study [76, 15].

We should note that there exists a myriad of techniques, including metric learning, kernel design, autoencoders, and others [77, 78, 79, 80, 81] to build feature spaces and perform unsupervised learning. Many of these works share a similar goal to ours, and we opted for SSL due to its proven efficacy in fields like computer vision and the direct analogy offered by data augmentations. One particular methodology that deserves mention is that of multi-fidelity modeling, which can reduce dependency on extensive training data for learning physical tasks [82, 83, 84]. The goals of multi-fidelity modeling include training with data of different fidelity [82] or enhancing the accuracy of models by incorporating high quality data into models [85]. In contrast, SSL aims to harness salient features from diverse data sources without being tailored to specific applications. The techniques we employ capitalize on the inherent structure in a dataset, especially through augmentations and invariances.

**Equivariant networks and geometric deep learning** In the past several years, an extensive set of literature has explored questions in the so-called realm of geometric deep learning tying together aspects of group theory, geometry, and deep learning [86]. In one line of work, networks have been designed to explicitly encode symmetries into the network via equivariant layers or explicitly symmetric parameterizations [87, 88, 89, 90]. These techniques have notably found particular application in chemistry and biology related problems [91, 92, 93] as well as learning on graphs [94]. Another line of work considers optimization over layers or networks that are parameterized over a Lie group [95, 96, 97, 98, 99]. Our work does not explicitly encode invariances or structurally parameterize Lie groups into architectures as in many of these works, but instead tries to learn representations that are approximately symmetric and invariant to these group structures via the SSL. As mentioned in the main text, perhaps more relevant for future work are techniques for learning equivariant features and maps [41, 42].

## E  Details on Augmentations

The generators of the Lie point symmetries of the various equations we study are listed below. For symmetry augmentations which distort the periodic grid in space and time, we provide inputs $x$ and $t$ to the network which contain the new spatial and time coordinates after augmentation.

### E.1  Burgers' equation

As a reminder, the Burgers' equation takes the form

$$u_t + uu_x - \nu u_{xx} = 0. \tag{49}$$

Lie point symmetries of the Burgers' equation are listed in Table 4. There are five generators. As we will see, the first three generators corresponding to translations and Galilean boosts are consistent with the other equations we study (KS, KdV, and Navier Stokes) as these are all flow equations.

Table 4: Generators of the Lie point symmetry group of the Burgers' equation in its standard form [44, 100].

| | Lie algebra generator | Group operation $(x, t, u) \mapsto$ |
|---|---|---|
| $g_1$ (space translation) | $\epsilon \partial_x$ | $(\,x + \epsilon\,, t, u)$ |
| $g_2$ (time translation) | $\epsilon \partial_t$ | $(x,\, t + \epsilon\,, u)$ |
| $g_3$ (Galilean boost) | $\epsilon(t\partial_x + \partial_u)$ | $(\,x + \epsilon t\,, t,\, u + \epsilon\,)$ |
| $g_4$ (scaling) | $\epsilon(x\partial_x + 2t\partial_t - u\partial_u)$ | $(\,e^\epsilon x\,,\, e^{2\epsilon}t\,,\, e^{-\epsilon}u\,)$ |
| $g_5$ (projective) | $\epsilon(xt\partial_x + t^2\partial_t + (x - tu)\partial_u)$ | $\left( \dfrac{x}{1 - \epsilon t}\,,\, \dfrac{t}{1 - \epsilon t}\,,\, u + \epsilon(x - tu) \right)$ |

**Comments regarding error in [12]**  As a cautionary note, the symmetry group given in Table 1 of [12] for Burgers' equation is incorrectly labeled for Burgers' equation in its *standard* form. Instead, these augmentations are those for Burgers' equation in its *potential* form, which is given as:

$$u_t + \frac{1}{2}u_x^2 - \nu u_{xx} = 0. \tag{50}$$

Burgers' equation in its standard form is $v_t + vv_x - \nu v_{xx} = 0$, which can be obtained from the transformation $v = u_x$. The Lie point symmetry group of the equation in its potential form contains more generators than that of the standard form. To apply these generators to the standard form of Burgers' equation, one can convert them via the Cole-Hopf transformation, but this conversion loses the smoothness and locality of some of these transformations (i.e., some are no longer Lie point transformations, although they do still describe valid transformations between solutions of the equation's corresponding form).

Note that this discrepancy does not carry through in their experiments: [12] only consider input data as solutions to Heat equation, which they subsequently transform into solutions of Burgers' equation via a Cole-Hopf transform. Therefore, in their code, they apply augmentations using the Heat equation, for which they have the correct symmetry group. We opted only to work with solutions to Burgers' equations itself for a slightly fairer comparison to real-world settings, where a convenient transform to a linear PDE such as the Cole-Hopf transform is generally not available.

### E.2  KdV

Lie point symmetries of the KdV equation are listed in Table 5. Though all the operations listed are valid generators of the symmetry group, only $g_1$ and $g_3$ are invariant to the downstream task of the inverse problem. (Notably, these parameters are independent of any spatial shift). Consequently, during SSL pre-training for the inverse problem, only $g_1$ and $g_3$ were used for learning representations. In contrast, for time-stepping, all listed symmetry groups were used.

Table 5: Generators of the Lie point symmetry group of the KdV equation. The only symmetries used in the inverse task of predicting initial conditions are $g_1$ and $g_3$ since the other two are not invariant to the downstream task.

| | Lie algebra generator | Group operation $(x, t, u) \mapsto$ |
|---|---|---|
| $g_1$ (space translation) | $\epsilon \partial_x$ | $(\,x + \epsilon\,, t, u)$ |
| $g_2$ (time translation) | $\epsilon \partial_t$ | $(x,\, t + \epsilon\,, u)$ |
| $g_3$ (Galilean boost) | $\epsilon(t\partial_x + \partial_u)$ | $(\,x + \epsilon t\,, t,\, u + \epsilon\,)$ |
| $g_4$ (scaling) | $\epsilon(x\partial_x + 3t\partial_t - 2u\partial_u)$ | $(\,e^\epsilon x\,,\, e^{3\epsilon}t\,,\, e^{-2\epsilon}u\,)$ |

### E.3 KS

Lie point symmetries of the KS equation are listed in Table 6. All of these symmetry generators are shared with the KdV equation listed in Table 4. Similar to KdV, only $g_1$ and $g_3$ are invariant to the downstream regression task of predicting the initial conditions. In addition, for time-stepping, all symmetry groups were used in learning meaningful representations.

Table 6: Generators of the Lie point symmetry group of the KS equation. The only symmetries used in the inverse task of predicting initial conditions are $g_1$ and $g_3$ since $g_2$ is not invariant to the downstream task.

|  | Lie algebra generator | Group operation $(x, t, u) \mapsto$ |
| --- | --- | --- |
| $g_1$ (space translation) | $\epsilon \partial_x$ | $(\,x + \epsilon\,, t, u)$ |
| $g_2$ (time translation) | $\epsilon \partial_t$ | $(x,\, t + \epsilon\,, u)$ |
| $g_3$ (Galilean boost) | $\epsilon(t \partial_x + \partial_u)$ | $(\,x + \epsilon t\,, t,\, u + \epsilon\,)$ |

### E.4 Navier Stokes

Lie point symmetries of the incompressible Navier Stokes equation are listed in Table 7 [101]. As pressure is not given as an input to any of our networks, the symmetry $g_q$ was not included in our implementations. For augmentations $g_{E_x}$ and $g_{E_y}$, we restricted attention only to linear $E_x(t) = E_y(t) = t$ or quadratic $E_x(t) = E_y(t) = t^2$ functions. This restriction was made to maintain invariance to the downstream task of buoyancy force prediction in the linear case or easily calculable perturbations to the buoyancy by an amount $2\epsilon$ to the magnitude in the quadratic case. Finally, we fix both order and steps parameters in our Lie-Trotter approximation implementation to 2 for computationnal efficiency.

## F    Experimental details

Whereas we implemented our own pretraining and evaluation (kinematic viscosity, initial conditions and buoyancy) pipelines, we used the data generation and time-stepping code provided on Github by [12] for Burgers', KS and KdV, and in [18] for Navier-Stokes (MIT License), with slight modification to condition the neural operators on our representation. All our code relies relies on Pytorch. Note that the time-stepping code for Navier-Stokes uses Pytorch Lightning. We report the details of the training cost and hyperparameters for pretraining and timestepping in Table 9 and Table 10 respectively.

### F.1    Experiments on Burgers' Equation

Solutions realizations of Burgers' equation were generated using the analytical solution [32] obtained from the Heat equation and the Cole-Hopf transform. During generation, kinematic viscosities, $\nu$, and initial conditions were varied.

**Representation pretraining**    We pretrain a representation on subsets of our full dataset containing $10,000$ 1D time evolutions from Burgers equation with various kinematic viscosities, $\nu$, sampled uniformly in the range $[0.001, 0.007]$, and initial conditions using a similar procedure to [12]. We generate solutions of size $224 \times 448$ in the spatial and temporal dimensions respectively, using the default parameters from [12]. We train a ResNet18 [17] encoder using the VICReg [9] approach to joint embedding SSL, with a smaller projector (width $512$) since we use a smaller ResNet than in the original paper. We keep the same variance, invariance and covariance parameters as in [9]. We use the following augmentations and strengths:

- Crop of size $(128, 256)$, respectively, in the spatial and temporal dimension.
- Uniform sampling in $[-2, 2]$ for the coefficient associated to $g_1$.
- Uniform sampling in $[0, 2]$ for the coefficient associated to $g_2$.
- Uniform sampling in $[-0.2, 0.2]$ for the coefficient associated to $g_3$.

Table 7: Generators of the Lie point symmetry group of the incompressible Navier Stokes equation. Here, $u, v$ correspond to the velocity of the fluid in the $x, y$ direction respectively and $p$ corresponds to the pressure. The last three augmentations correspond to infinite dimensional Lie subgroups with choice of functions $E_x(t), E_y(t), q(t)$ that depend on $t$ only. For invariant tasks, we only used settings where $E_x(t), E_y(t) = t$ (linear) or $E_x(t), E_y(t) = t^2$ (quadratic) to ensure invariance to the downstream task or predictable changes in the outputs of the downstream task. These augmentations are listed as numbers 6 to 9.

| | Lie algebra generator | Group operation $(x, y, t, u, v, p) \mapsto$ |
|---|---|---|
| $g_1$ (time translation) | $\epsilon \partial_t$ | $(x, y, t+\epsilon, u, v, p)$ |
| $g_2$ ($x$ translation) | $\epsilon \partial_x$ | $(x+\epsilon, y, t, u, v, p)$ |
| $g_3$ ($y$ translation) | $\epsilon \partial_y$ | $(x, y+\epsilon, t, u, v, p)$ |
| $g_4$ (scaling) | $\epsilon(2t\partial_t + x\partial_x + y\partial_y \\ - u\partial_u - v\partial_v - 2p\partial_p)$ | $(e^\epsilon x, e^\epsilon y, e^{2\epsilon}t, e^{-\epsilon}u, e^{-\epsilon}v, e^{-2\epsilon}p)$ |
| $g_5$ (rotation) | $\epsilon(x\partial_y - y\partial_x + u\partial_v - v\partial_u)$ | $(x\cos\epsilon - y\sin\epsilon, x\sin\epsilon + y\cos\epsilon, t, \\ u\cos\epsilon - v\sin\epsilon, u\sin\epsilon + v\cos\epsilon, p)$ |
| $g_6$ ($x$ linear boost)[1] | $\epsilon(t\partial_x + \partial_u)$ | $(x+\epsilon t, y, t, u+\epsilon, v, p)$ |
| $g_7$ ($y$ linear boost)[1] | $\epsilon(t\partial_y + \partial_v)$ | $(x, y+\epsilon t, t, u, v+\epsilon, p)$ |
| $g_8$ ($x$ quadratic boost)[2] | $\epsilon(t^2\partial_x + 2t\partial_u - 2x\partial_p)$ | $(x+\epsilon t^2, y, t, u+2\epsilon t, v, p-2x)$ |
| $g_9$ ($y$ quadratic boost)[2] | $\epsilon(t^2\partial_y + 2t\partial_v - 2y\partial_p)$ | $(x, y+\epsilon t^2, t, u, v+2\epsilon t, p-2y)$ |
| $g_{E_x}$ ($x$ general boost)[3] | $\epsilon(E_x(t)\partial_x + E'_x(t)\partial_u \\ - xE''_x(t)\partial_p)$ | $(x+\epsilon E_x(t), y, t, \\ u+\epsilon E'_x(t), v, p-E''x(t)x)$ |
| $g_{E_y}$ ($y$ general boost)[3] | $\epsilon(E_y(t)\partial_y + E'y(t)\partial_v \\ - yE''y(t)\partial_p)$ | $(x, y+\epsilon E_y(t), t, \\ u, v+\epsilon E'y(t), p-E''y(t)y)$ |
| $g_q$ (additive pressure)[3] | $\epsilon q(t)\partial_p$ | $(x, y, t, u, v, p+q(t))$ |

[1] case of $g_{E_x}$ or $g_{E_y}$ where $E_x(t) = E_y(t) = t$ (linear function of $t$)
[2] case of $g_{E_x}$ or $g_{E_y}$ where $E_x(t) = E_y(t) = t^2$ (quadratic function of $t$)
[3] $E_x(t), E_y(t), q(t)$ can be any given smooth function that only depends on $t$

- Uniform sampling in $[-1, 1]$ for the coefficient associated to $g_4$.

We pretrain for 100 epochs using AdamW [33] and a batch size of 32. Crucially, we assess the quality of the learned representation via linear probing for kinematic viscosity regression, which we detail below.

**Kinematic viscosity regression** We evaluate the learned representation as follows: the ResNet18 is frozen and used as an encoder to produce features from the training dataset. The features are passed through a linear layer, followed by a sigmoid to constrain the output within $[\nu_{\min}, \nu_{\max}]$. The learned model is evaluated against our validation dataset, which is comprised of $2,000$ samples.

**Time-stepping** We use a 1D CNN solver from [12] as our baseline. This neural solver takes $T_p$ previous time steps as input, to predict the next $T_f$ future ones. Each channel (or spatial axis, if we view the input as a 2D image with one channel) is composed of the realization values, $u$, at $T_p$ times, with spatial step size $dx$, and time step size $dt$. The dimension of the input is therefore $(T_p + 2, 224)$, where the extra two dimensions are simply to capture the scalars $dx$ and $dt$. We augment this input with our representation. More precisely, we select the encoder that allows for the most accurate linear regression of $\nu$ with our validation dataset, feed it with the CNN operator input and reduce the resulting representation dimension to $d$ with a learned projection before adding it as supplementary channels to the input, which is now $(T_p + 2 + d, 224)$.

We set $T_p = 20$, $T_f = 20$, and $n_{\text{samples}} = 2,000$. We train both models for 20 epochs fol-

Table 8: One-step validation NMSE for time-stepping on Burgers for different architectures.

| Architecture | ResNet1d | FNO1d |
|---|---|---|
| Baseline (no conditioning) | $0.110 \pm 0.008$ | $0.184 \pm 0.002$ |
| Representation conditioning | $\mathbf{0.108 \pm 0.011}$ | $\mathbf{0.173 \pm 0.002}$ |

lowing the setup from [12]. In addition, we use AdamW with a decaying learning rate and different configurations of 3 runs each:

- Batch size $\in \{16, 64\}$.
- Learning rate $\in \{0.0001, 0.00005\}$.

## F.2 Experiments on KdV and KS

To obtain realizations of both the KdV and KS PDEs, we apply the method of lines, and compute spatial derivatives using a pseudo-spectral method, in line with the approach taken by [12].

**Representation pretraining** To train on realizations of KdV, we use the following VICReg parameters: $\lambda_{var} = 25$, $\lambda_{inv} = 25$, and $\lambda_{cov} = 4$. For the KS PDE, the $\lambda_{var}$ and $\lambda_{inv}$ remain unchanged, with $\lambda_{cov} = 6$. The pre-training is performed on a dataset comprised of $10,000$ 1D time evolutions of each PDE, each generated from initial conditions described in the main text. Generated solutions were of size $128 \times 256$ in the spatial and temporal dimensions, respectively. Similar to Burgers' equation, a ResNet18 encoder in conjunction with a projector of width $512$ was used for SSL pre-training. The following augmentations and strengths were applied:

- Crop of size $(32, 256)$, respectively, in the spatial and temporal dimension.
- Uniform sampling in $[-0.2, 0.2]$ for the coefficient associated to $g_3$.

**Initial condition regression** The quality of the learned representations is evaluated by freezing the ResNet18 encoder, training a separate regression head to predict values of $A_k$ and $\omega_k$, and comparing the NMSE to a supervised baseline. The regression head was a fully-connected network, where the output dimension is commensurate with the number of initial conditions used. In addition, a range-constrained sigmoid was added to bound the output between $[-0.5, 2\pi]$, where the bounds were informed by the minimum and maximum range of the sampled initial conditions. Lastly, similar to Burgers' equation, the validation dataset is comprised of $2,000$ labeled samples.

**Time-stepping** The same 1D CNN solver used for Burgers' equation serves as the baseline for time-stepping the KdV and KS PDEs. We select the ResNet18 encoder based on the one that provides the most accurate predictions of the initial conditions with our validation set. Here, the input dimension is now $(T_p + 2, 128)$ to agree with the size of the generated input data. Similarly to Burgers' equation, $T_p = 20$, $T_f = 20$, and $n_{\text{samples}} = 2,000$. Lastly, AdamW with the same learning rate and batch size configurations as those seen for Burgers' equation were used across 3 time-stepping runs each.

A sample visualization with predicted instances of the KdV PDE is provided in Fig. 7 below:

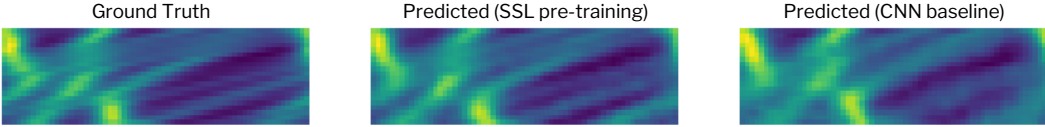

Figure 7: Illustration of the 20 predicted time steps for the KdV PDE. (**Left**) Ground truth data from PDE solver; (**Middle**) Predicted $u(x, t)$ using learned representations; (**Right**) Predicted output from using the CNN baseline.

Table 9: List of model hyperparameters and training details for the invariant tasks. Training time includes periodic evaluations during the pretraining.

| Equation | Burgers' | KdV | KS | Navier Stokes |
|---|---|---|---|---|
| *Network:* | | | | |
| Model | ResNet18 | ResNet18 | ResNet18 | ResNet18 |
| Embedding Dim. | 512 | 512 | 512 | 512 |
| *Optimization:* | | | | |
| Optimizer | LARS [102] | AdamW | AdamW | AdamW |
| Learning Rate | 0.6 | 0.3 | 0.3 | 3e-4 |
| Batch Size | 32 | 64 | 64 | 64 |
| Epochs | 100 | 100 | 100 | 100 |
| Nb of exps | $\sim 300$ | $\sim 30$ | $\sim 30$ | $\sim 300$ |
| *Hardware:* | | | | |
| GPU used | Nvidia V100 | Nvidia M4000 | Nvidia M4000 | Nvidia V100 |
| Training time | $\sim 5h$ | $\sim 11h$ | $\sim 12h$ | $\sim 48h$ |

### F.3   Experiments on Navier-Stokes

We use the Conditioning dataset for Navier Stokes-2D proposed in [18], consisting of 26,624 2D time evolutions with 56 time steps and various buoyancies ranging approximately uniformly from 0.2 to 0.5.

**Representation pretraining**   We train a ResNet18 for 100 epochs with AdamW, a batch size of 64 and a learning rate of 3e-4. We use the same VICReg hyperparameters as for Burgers' Equation. We use the following augmentations and strengths (augmentations whose strength is not specified here are not used):

- Crop of size $(16, 128, 128)$, respectively in temporal, x and y dimensions.
- Uniform sampling in $[-1, 1]$ for the coefficients associated to $g_2$ and $g_3$ (applied respectively in x and y).
- Uniform sampling in $[-0.1, 0.1]$ for the coefficients associated to $g_5$.
- Uniform sampling in $[-0.01, 0.01]$ for the coefficients associated to $g_6$ and $g_7$ (applied respectively in x and y).
- Uniform sampling in $[-0.01, 0.01]$ for the coefficients associated to $g_8$ and $g_9$ (applied respectively in x and y).

**Buoyancy regression**   We evaluate the learned representation as follows: the ResNet18 is frozen and used as an encoder to produce features from the training dataset. The features are passed through a linear layer, followed by a sigmoid to constrain the output within $[\text{Buoyancy}_{min}, \text{Buoyancy}_{max}]$. Both the fully supervised baseline (ResNet18 + linear head) and our (frozen ResNet18 + linear head) model are trained on $3,328$ unseen samples and evaluated against $6,592$ unseen samples.

**Time-stepping**   We mainly depart from [18] by using 20 epochs to learn from 1,664 trajectories as we observe the results to be similar, and allowing to explore more combinations of architectures and conditioning methods.

**Time-stepping results**   In addition to results on 1,664 trajectories, we also perform experiments with bigger train dataset (6,656) as in [18], using 20 epochs instead of 50 for computational reasons. We also report results for the two different conditioning methods described in [18], Addition and AdaGN. The results can be found in Table 11. As in [18], AdaGN outperforms Addition. Note that AdaGN is needed for our representation conditioning to significantly improve over no conditioning. Finally, we found a very small bottleneck in the MLP that process the representation to also be crucial for performance, with a size of 1 giving the best results.

Table 10: List of model hyperparameters and training details for the timestepping tasks.

| Equation | Burgers' | KdV | KS | Navier Stokes |
|---|---|---|---|---|
| *Neural Operator:* | | | | |
| Model | CNN [12] | CNN [12] | CNN [12] | Modified U-Net-64 [18] |
| *Optimization:* | | | | |
| Optimizer | AdamW | AdamW | AdamW | Adam |
| Learning Rate | 1e-4 | 1e-4 | 1e-4 | 2e-4 |
| Batch Size | 16 | 16 | 16 | 32 |
| Epochs | 20 | 20 | 20 | 20 |
| *Hardware:* | | | | |
| GPU used | Nvidia V100 | Nvidia M4000 | Nvidia M4000 | Nvidia V100 (16) |
| Training time | $\sim 1d$ | $\sim 2d$ | $\sim 2d$ | $\sim 1.5d$ |

Table 11: One-step validation MSE $\times 1e^{-3}$ ($\downarrow$) for Navier-Stokes for different baselines and conditioning methods, with UNet$_{\text{mod64}}$ [18] as base model.

| Dataset size | 1,664 | 6,656 |
|---|---|---|
| *Methods without ground truth buoyancy:* | | |
| Time conditioned, Addition | $2.60 \pm 0.05$ | $1.18 \pm 0.03$ |
| Time + Rep. conditioned, Addition (ours) | $2.47 \pm 0.02$ | $1.17 \pm 0.04$ |
| Time conditioned, AdaGN | $2.37 \pm 0.01$ | $1.12 \pm 0.02$ |
| Time + Rep. conditioned, AdaGN (ours) | $\mathbf{2.35 \pm 0.03}$ | $\mathbf{1.11 \pm 0.01}$ |
| *Methods with ground truth buoyancy:* | | |
| Time + Buoyancy conditioned, Addition | $2.08 \pm 0.02$ | $1.10 \pm 0.01$ |
| Time + Buoyancy conditioned, AdaGN | $\mathbf{2.01 \pm 0.02}$ | $\mathbf{1.06 \pm 0.04}$ |

