# Table of Contents

**Typo** There is a typo in Figure 5: $g_1$ is translation applied in t (not x), while $g_2$ is translation applied in x (not y). Whenever applicable, we use the same strength in both x and y axis.

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

 use smaller trajectories (32) as in [18] (56) to reduce computational burden. To condition on our representation, we simply replace the Fourier embedding of the buoyancy by a learned projection of our representation. We compare our conditioning to the parameter conditioning, and no conditioning. All methods are however conditioned on time, and use a single frame to predict a future one. We use the same base configuration as the one provided in [18] for conditioning with modified UNet-64, except we double the effective batch size (since we use 8 GPUs instead of 4) and thus increase the base learning rate to 1e-3. We also depart from [18] by evaluating the learned PDE surrogate at four subsequent time horizons: $\{1, 2, 4, 8\}$.

Table 9: Time-stepping MSE (↓) for Navier-Stokes on various time horizons.

| Time horizon | 1 | 2 | 4 | 8 |
|---|---|---|---|---|
| *Method:* | | | | |
| Time conditioned | $0.0028 \pm 0.0001$ | $0.0035 \pm 0.0001$ | $0.0053 \pm 0.0001$ | $0.0106 \pm 0.0001$ |
| Time + Rep. cond. (ours) | $0.0008 \pm 0.0001$ | $0.0014 \pm 0.0001$ | $0.0032 \pm 0.0001$ | $0.0092 \pm 0.0001$ |
| Time + Param. cond. | $0.0006 \pm 0.0001$ | $0.0013 \pm 0.0001$ | $0.0027 \pm 0.0001$ | $0.0091 \pm 0.0001$ |

**Time-stepping results.** We report our complete results after 20k iterations in Table 9.

In order for the appendix to be self-contained, we include references again at the end of the appendix. This reference numbering includes references that appear in the appendix, but not the main body of the paper.