# OpenReview forum: "Self-Supervised Learning with Lie Symmetries for Partial Differential Equations"
_NeurIPS.cc/2023/Conference — NeurIPS 2023 poster_

### Official Review · Reviewer_rnp9 · 2023-06-09

**Soundness:** 4 excellent
**Presentation:** 4 excellent
**Contribution:** 3 good
**Rating:** 8
**Confidence:** 4

**Summary:**

The paper presents an approach for self-supervised learning of PDEs. The main approach uses Lie symmetries (similar e.g. to translational symmetries for images) in the solutions of differential equations not only for data augmentation (as has been done before), but for representation learning for downsteam tasks.

**Strengths:**

The paper is a pleasure to read. There are nice diagrams (perhaps even too fancy?) and a nice verbal presentation everywhere. The appendix comes with an introduction into the theory, which is also well written and correct. The literature is thoroughly cited (Of course, there is always more to cite, but the amount is huge and the choice are reasonable.) Many experiments are conducted. Even limitations and drawbacks are clearly stated, such that after reading the paper a really feel what works easily, what works with problems, and what is not yet known. I feel well-informed after reading the paper.

Furthermore, the topic is clearly relevant for NeurIPS: it is a non-trivial machine learning question with relevant application domains.

In particular the correct usage of PDE-terms is refreshing to see when reviewing NeurIPS papers about PDEs. Here, I could find no problems. All terms were used correctly, even though more could and should have been said about PDEs, but the paper only has 9 pages; more is saind in the appendix. This quality in mathematical writing is not the norm at NeurIPS! And, at the same time, the paper is not technical.


**Weaknesses:**

I could follow the paper. However, I have a strong math background that includes symmetries of PDEs. The authors try really hard (and are quite good) in making everything clear to the audience. However, I guess that much of the NeurIPS audience might have problems with some techniques used in the paper.

After reading the paper, it was not clear to me what kind of data is necessary. In one of your experiments, you used data on a grid and were thus able to apply ResNet18. More details in this regard (what is possible? what is reasonable?) would have been nice.

Quite a lot of data is necessary for training. If I have 10000 samples of a PDE, I kind of know the PDE anyway. This, together with the high dependency on several hyperparameters and the additional limitation (see below), is the main problem of this paper which make is "just a strong NeurIPS paper" instead of an outstanding paper.

**Questions:**

I guess for a paper of this level ob abstraction, you do not need to define what a group and what a group operation is? (l. 115)

**Limitations:**

The approach is certainly interesting. However, where I am working with differential equations, one only knows the symmetries from the differential equations. Hence, I rarely see a need for this approach.

Otherwise, limitations and open questions are given both in the paper (e.g. l. 99ff, 130ff, etc.)

---

> ### Author Rebuttal · Authors · 2023-08-10
>
> We thank the reviewer for taking their time to review the paper and providing valuable comments and feedback. We are glad the reviewer is happy with the quality of the experiments and the completeness of our manuscript. We address the reviewer’s questions and comments below.
>
> ***
>
> ### Question 1:
> >After reading the paper, it was not clear to me what kind of data is necessary. In one of your experiments, you used data on a grid and were thus able to apply ResNet18. More details in this regard (what is possible? what is reasonable?) would have been nice.
>
> Thank you for raising this interesting point. Going further, real-world data could indeed come in the form of irregular grids making the use of architectures such as ResNet more challenging. There, one could leverage neural networks for solving PDEs with graph based methods that handle irregular grids [1-2] and SSL frameworks that have been extended to architectures more suited to such data [3]. Additionally, SSL for computer vision comes with recipes for using transformers, which could also be leveraged for irregular grids although such an approach may require more data than ResNets or GNNs. We will add discussion around these points to the updated draft of our paper.
>
> As an aside, Lie symmetries are still implementable on irregular grids, since implementing Lie point symmetries only requires knowing the values of the solution at any given point (hence the name). This is not necessarily true of other types of symmetries such as Lie Bäcklund symmetries which require knowing the solution at a neighborhood around every point to implement them.
>
> ### Question 2:
> >Quite a lot of data is necessary for training. If I have 10000 samples of a PDE, I kind of know the PDE anyway. This, together with the high dependency on several hyperparameters and the additional limitation (see below), is the main problem of this paper which make is "just a strong NeurIPS paper" instead of an outstanding paper.
>
> Although our current approach assumes we know the family of the PDE, our datasets mix realizations that have different initial conditions and equation parameters such as kinematic viscosity or buoyancy. In that regard, having models that, as those we propose, generalize to unseen initial conditions or equation parameters is already very valuable.
>
> We also refer the reviewer to our answer to reviewer swdq, where we provide an experiment learning representations from a dataset mixing Burgers, KdV and KS realizations. Although very preliminary, this setup would remove the need to know the PDE family, while alleviating the data requirement for each equation.
>
> We also show in figure 4 that we already have improved performance with our approach with only a few thousands data points. While this may already be a big requirement in certain scenarios, this indicates that it can still be applied in more data-constrained regimes.
>
> ### Other relevant comment:
>
> > The approach is certainly interesting. However, where I am working with differential equations, one only knows the symmetries from the differential equations.
>
> Thank you for pointing this out. Indeed, knowing the differential equation allows one to derive symmetries. This is the most ideal setting. Nevertheless, symmetries can also be deduced by knowing what class of differential equations one PDE is a part of or knowing properties of the PDE. For example, all flow related equations share common symmetries such as translations and Galilean boosts (e.g., see the added shared experiment in response to reviewer swdq). Furthermore, many symmetries can be derived from known conservation properties by Noether's theorems. So in general, knowing an equation precisely is best, but this is not always required in practice. Thank you again for raising this clarification. We will comment on this further in our updated draft.
>
> **References:**
>
> [1] Belbute-Peres, Filipe De Avila, Thomas Economon, and Zico Kolter. "Combining differentiable PDE solvers and graph neural networks for fluid flow prediction." international conference on machine learning. PMLR, 2020.
>
> [2] Brandstetter, Johannes, Daniel Worrall, and Max Welling. "Message passing neural PDE solvers." arXiv preprint arXiv:2202.03376 (2022).
>
> [3] You, Yuning et al., “Graph Contrastive Learning with Augmentations”, NeurIPS 2020

---

> ### Comment · Reviewer_rnp9 · 2023-08-10
> **Opinion after the rebuttals**
>
> I have read all reviews and all rebuttals. Having given the most positive rating, I will try to justify my unchanged opinion.
>
>  - All reviewers agree that the paper is well-written.
>  - All reviewers agree that the method is (mostly) correct.
>  - The main open point regarding correctness is the boundary conditions. In my corner of differential equations, the boundary conditions are more like additional information rather than an integral part of the differential equations. Hence, at least in my corner of differential equations, the raised problem is less problematic.
>  - The primary point of difference is the amount of novelty. Since I feel that differential equations, especially partial differential equations, pose really challenging questions, the level of novelty I see in this paper is sufficient for me to consider it a valuable contribution to NeurIPS. I acknowledge that this point is highly subjective.

---

### Official Review · Reviewer_xeoP · 2023-07-06

**Soundness:** 2 fair
**Presentation:** 3 good
**Contribution:** 1 poor
**Rating:** 3
**Confidence:** 4

**Summary:**

This paper proposes to learn general-purpose representations of PDEs from heterogeneous  data by implementing joint embedding methods for self-supervised learning. Learned representation outperforms baseline approaches for invariant tasks such as regressing the coefficients of a PDE and improve the time-stepping performance of neural solvers.

**Strengths:**

- Propose a general framework for self-supervised learning in PDE by using symmetry transformation in PDE.
- Learned representation can be used in several tasks, such as determining unknown parameters and time-stepping.
- This paper is well-written and the presentation is clear.


**Weaknesses:**

- The novelty of this paper is quite marginal. The SSL framework adopted in the paper is well-known and not customized for the specific PDE problem. The only contribution is the augmentation of PDE solutions according to symmetry groups, which is also well-studied in previous literature, such as in [1].
- The evaluation on time-stepping is obviously not enough. For example, this paper doesn’t compare with several important baselines, such as MPPDE [2], FNO [3], UNet, PINN, etc. I think the authors need to show the effectiveness of using learned representation to improve at least a few of these models on limited labeled training data.
- The regression task of determining external buoyancy force (a constant value) in the NS equation is quite simple in practice. In practice, there are more complicated forcing terms. For example, in FNO [3], they use a forcing term containing cosine and sine waves. Can authors use the forcing term in FNO and regress the parameters in that forcing term?
- The viscosity in the NS equation is also an important parameter. Can authors also provide regression results on determining the viscosity?

[1] Brandstetter, Johannes, Max Welling, and Daniel E. Worrall. "Lie point symmetry data augmentation for neural PDE solvers." International Conference on Machine Learning. PMLR, 2022.

[2] Brandstetter, Johannes, Daniel Worrall, and Max Welling. "Message passing neural PDE solvers." arXiv preprint arXiv:2202.03376 (2022).

[3] Li, Zongyi, et al. "Fourier neural operator for parametric partial differential equations." arXiv preprint arXiv:2010.08895 (2020).

**Questions:**

- Why do authors only consider Lie point symmetry for PDE? Is there any other symmetry group that applies to PDE?
- If a PDE does not have periodic boundary conditions, does augmentation using symmetry group still valid?
- Can elaborate more on why learned representation cannot improve time-stepping performance for the Burgers equation?


**Limitations:**

The limitations of this paper are discussed.

---

> ### Author Rebuttal · Authors · 2023-08-10
>
> We thank the reviewer for their feedback and insightful comments. We address the reviewer’s questions below and at times, challenge their criticisms. We welcome further discussion.
>
> ***
> ### Novelty
>
> We acknowledge that SSL for computer vision and the study of symmetry groups of PDEs are *separately* well-established. We respectfully disagree on novelty, and share examples that shed light on our contributions:
> - The role of data augmentation in supervised learning [1] and SSL varies fundamentally. In SSL, augmentations dictate properties preserved by the encoder, and no learning happens without them. As further illustration of this, our conclusions differ from [1] on the utility of augmentations. For instance, time translations often impede learning good representations for Burgers' or Navier-Stokes, contrasting with findings in [1] (Tables 3d and 4d).
> - The Lie-Trotter techniques for implementing augmentations (Appendix B) are new to the ML for PDE community as far as we are aware, and enable more smoothly and universally applying augmentations. This implementation may even be useful outside of SSL. For comparison Ref. [1] applied Lie symmetries by choosing an arbitrary order for the basis elements, which neither guarantee universality (there exist augmentations which cannot be performed) nor smoothness (the Lie algebra has no guarantees on norm of augmentation).
> - SSL comes with many moving parts to which subsequent representations may or may not be sensitive [2]. For example, it was unclear a priori that using the generic task of coefficient or initial condition regression as metrics for pre-training evaluation (replacing the ImageNet top-1 accuracy in computer vision) would select embeddings that transfer nicely to other tasks such as time-stepping.
>
> SSL algorithms based on joint embedding architectures have application in various regimes outside its original use in computer vision, including published works for graphs [3], language [4], point clouds [5], and more. These works elucidate where SSL is useful, how it can be adapted, and the extent to which SSL can be applied for learning real-world data in general. Overall, we believe it is very valuable to share our insights gained for PDE data.
>
>
> ### Improving other models such as FNO
>
> We acknowledge that using our representations to improve other baselines would make this work stronger. We refer the reviewer to our general comment, where we show the effectiveness of the SSL representations to condition FNOs, Fourier U-Nets, and U-Nets with a supplementary conditioning method, AdaGN.
>
> ### Determining external buoyancy in the NS equation
>
> From a certain viewpoint, regressing the buoyancy force (a constant), within the NS equation might appear straightforward. One can simply calculate the derivatives and functions in the PDE and regress it. However, this is not the case here:
> Crucially, our network is agnostic to the specific form of the PDE. To regress the buoyancy force, it has to disentangle the parameter from the complex, nonlinear map of the PDE.
> Even in a supervised setting, the performance is far from the buoyancy's resolution, showing that it remains a hard task empirically.
>
> This makes our task valid to gauge the quality of our representation.
>
> ### Navier-Stokes regression results on viscosity
>
> The NS benchmark we use (PDEArena) comes with a fixed viscosity. We did not have sufficient time to generate a new dataset with varying viscosities.  However, we will add this task in a revised version of the manuscript.
>
> ### On the choice of Lie point symmetry for PDEs
> We primarily focused on Lie point symmetries due to their systematic derivation and well-established applications in PDEs. We acknowledge that other forms of symmetries, such as approximate, Bäcklund, and discrete transformations, can also be considered. However, these are typically challenging to derive, may introduce other sources of errors and, like Lie symmetries, are typically derived in infinite domain settings (so do not address boundary issues). When learning data from multiple different types of PDEs, these may be useful but this would require different implementations and changes to our setup.
>
> ### On augmentation using symmetry groups in non-periodic boundary conditions
> You have touched upon an important limitation that we also point out in our work. Since Lie symmetries do not preserve boundaries, we limit and test the magnitude of transformations to avoid large errors (see discussion beginning on line 142). In short, addressing boundary conditions more directly would either escalate the complexity of the task or divert us from the generality we aimed to maintain. Please refer to the general response and response to reviewer VTPv for a more complete answer.
>
> ### On conditioning time-stepping for Burgers equation
> It is likely that the dynamics of Burgers’ equation are easy to predict with little room for improvement compared to KdV and KS given the same resources. A similar observation is made in [1] (Appendix D, Tables 3 and 4), where the normalized MSE for Burgers is far lower than that for KdV and KS, when provided with a few hundreds of samples.
>
>
> **References:**
>
> [1] Brandstetter, Johannes et al. "Lie point symmetry data augmentation for neural PDE solvers." ICML 2022.
>
> [2] Balestriero, Randall, et al. "A cookbook of self-supervised learning." arXiv preprint (2023).
>
> [3] Xie, Yaochen, et al. "Self-supervised learning of graph neural networks: A unified review." IEEE TPAMI (2022).
>
> [4] Chuang, Ching-Yao, et al. "Debiased contrastive learning." NeurIPS 2020.
>
> [5] Jiang, Li, et al. "Guided point contrastive learning for semi-supervised point cloud semantic segmentation." CVPR 2021.

---

### Official Review · Reviewer_swdq · 2023-07-07

**Soundness:** 3 good
**Presentation:** 3 good
**Contribution:** 3 good
**Rating:** 7
**Confidence:** 4

**Summary:**

In this paper, the authors propose to use self-supervised learning for obtaining an embedding that can robustly be used for predicting some quantities of interest or for time stepping. Particularly, they use the joint-embedding framework for SSL. They use symmetry groups for training the embedding to be invariant to these symmetry groups and thus capture the underlying behavior of the PDEs.

**Strengths:**

+ novelty in using Lie symmetry groups for SSL training.
+ Improvement in training physics-based models in much lesser time.
+ covered a comprehensive list of symmetry groups for different PDEs. This would be very useful to the community.

**Weaknesses:**

- For a person from an engineering background, I do think SSL is quite similar to the ideas of Multi-fidelity modeling (https://arxiv.org/abs/1609.07196, https://arxiv.org/abs/2110.04170, https://arxiv.org/abs/1903.00104). Both help reduce the training data and both are very easy to implement. Unfortunately, SSL for the quit nh
- The authors talk about taking steps for building foundation models, I would imagine some kind of amalgamation of all the PDEs while training the SSL. Like you use all the data from all the PDEs to build and train an SSL model. That seems closer to how foundation models would behave. I feel like the paper gives big hopes and doesn't deliver them in the implementation. If they were to not talk about foundation models and just say they build SSL frameworks for PDEs, I would have been very happy.
- The authors make a comparison between supervised ones and SSL-trained ones. but, there are a whole host of PDE-residual-based models like DeepONets, FNOsetc. (which the authors acknowledge in the supplement). It would be nice to know if SSL is even needed for PDE-based foundation models. Can PDE-residuals sufficiently help us navigate through foundation models? Is there a need for SSL? Thats not established fairly. (note that I acknowledge the comparisons, but is that sufficient to establish the necessity of SSL for PDEs?)

**Questions:**

See the weaknesses. Further, there are more questions that are not particularly weakening the paper but could make it clearer.

1. Figure 5 and the corresponding text is very interesting to me. It seems that the results from the Lie transformation(0.0038 compared to 0.0078 in supervised) and Crop (0.0052 compared to 0.0078) are not that different. It seems to me that applying some kind of SSL training is more important than using a particular Lie transformation per se. Is it worth the effort to find out the best possible selection of Lie point augmentations? Further, is this selection task dependent? How does one train a generic model for several tasks then?
2. It would be nice to see some additional information on the computational overhead from SSL compared to supervised training. (see weaknesses, ideally, I would like to also know the comparison with DeepONets)

---

> ### Author Rebuttal · Authors · 2023-08-10
>
> We thank the reviewer for taking their time to review the paper, praising the novelty of our work, and providing valuable comments and feedback. We address the reviewer’s questions and comments below.
>
> ***
>
> ### SSL and Multi-fidelity modeling
>
> Part of the reviewer’s response was cut off. Could they please clarify what remained in this comment?
>
> ### On foundation models for PDEs
>
> Indeed, we mention foundation models for PDEs as part of an aspirational research program, but only take the first step towards this goal. We will correct the phrasing accordingly. Moreover, to bring our work closer to this ambitious goal, we added new experiments training a common representation from the mixture of our Burgers, KdV and KS datasets (all models of chaotic flow that share many Lie point symmetries), using crops and experimenting with different sets of Lie augmentations that are common to all three.
>
> We evaluate the representations on parameter regression tasks and report averaged results over 3 runs. Our preliminary results are encouraging yet show that mixing data from different equations is not straightforward:
>
> | Task             | Burgers kin. viscosity (%)   |
> |-----------------------------|--------------|
> | Supervised  | 1.55 ± 0.01 |
> | SSL features |  1.53 ± 0.02|
>
> On Burgers we match the supervised baseline in a short training regimen (50 total epochs), showing that the model can learn good representations with heterogeneous data sources. However, in the short time frame of the rebuttal, we could not get conclusive results when evaluating on KS or KdV.
> Mixing the equations requires addressing new challenges (e.g dealing with different chaotic behavior and different time and length scales between PDEs). It is a very interesting direction for future work.
>
> ### Is SSL needed for foundation models? What about PDE-residuals?
>
> The reviewer raises some good points about the necessity of SSL, and if we understand correctly, their inquiries cover two related issues: (1) the comparative effectiveness of SSL in constructing foundation models, and (2) the requirement of SSL in light of PDE-residual-based methods, which can operate in semi-supervised or unsupervised environments.
>
>
> Addressing (1), to clarify: we are not asserting that SSL is categorically the superior approach for building foundation models or representation learning methods. Nevertheless, based on very strong results in the realm of image processing [5], we are confident in SSL's potential when applied to PDEs.
>
> For (2), we concur that many potentially effective PDE-residual-based methods exist. However, these methods seem to serve a different purpose, i.e learning an approximation to a differential operator, while SSL attempts to learn a rich yet easy to leverage feature space. Our added experiments with FNO (see summary response) show the latter can benefit the former. Could the reviewer elaborate on the comparison they expect?
>
> ### Selection of augmentations
>
> Using crops as the only augmentation provides a decent baseline. However, useful Lie point augmentations are required for best performance. We have evidence of this for both regression/classification tasks and time-stepping. We report two new experiments in this regard (see Table below):
> For Burgers’, we trained a representation with the crop augmentation only. It does not outperform the supervised baseline in kinematic viscosity regression as opposed to the representation trained with crop and Lie point augmentations.
>
> For Navier-Stokes, representations with MSE of 0.0052 (crop only) and 0.0038 (crop + Lie augmentations) reported in Fig. 5 for buoyancy regression are actually different. Our best time-stepping model (UNet + AdaGN conditioning) conditioned on the crop-only representation does not outperform the baseline, in contrast to conditioning on the representation trained with crop and Lie point augmentations.
>
> | Task             | Burgers kin. viscosity (%)   | Navier-Stokes time-stepping MSE (1e-3)   |
> |-----------------------------|--------------|-------------|
> | Supervised baseline  |  1.18 ± 0.07 | 2.37  ±  0.01 |
> | SSL features (Crop only) | 2.3 ± 0.2 | 2.9 ± 0.8 |
> | SSL features (Crop + Lie augmentations) | 0.97 ± 0.04  | 2.35 +- 0.03 |
>
> It may be possible to improve on a given task by `"overfitting” the augmentation parameters, but it is observed both in SSL for computer vision [5] and in our work that a good set of augmentations for SSL provides the best results across a wide range of tasks. In our experiments for example, the best representation evaluated during pre-training works out of the box for time-stepping (Table 1).
>
> ### Computational overhead of SSL vs supervised
>
> SSL pre-training typically has higher training costs compared to just supervised methods. Crucially, pre-training is a fixed cost, and subsequent training costs for downstream tasks starting from SSL features are negligible: one linear layer on top of frozen SSL features is usually sufficient, whereas supervised learning requires tuning the whole network to get similar results [1-2]. Since SSL features are computationally cheap to use and transfer better [3-4], the SSL approach is advantageous when features are reused. Since the purpose of SSL and DeepONets seem different (see comment above), could the reviewer elaborate on the comparison they expect?
>
> **References:**
>
> [1] Chen, Ting, et al. "A simple framework for contrastive learning of visual representations." International conference on machine learning. ICML 2020.
>
> [2] Bardes, Adrien et al. "Vicreg: Variance-invariance-covariance regularization for self-supervised learning." ICLR 2022.
>
> [3] Ericsson, Linus et al. "How well do self-supervised models transfer?." CVPR 2021.
>
> [4] Tian, Yonglong, et al. "Rethinking few-shot image classification: a good embedding is all you need?."ECCV 2020
>
> [5] Oquab, Maxime, et al. “DINOv2: learning robust visual features without supervision”, arXiv preprint (2023)

---

> > ### Comment · Reviewer_swdq · 2023-08-14
> >
> >
> > 1. Sorry, my previous comment was somehow cut off. I wanted to say that it would be unkind to ignore all the work on multi-fidelity modeling. The authors claim clearly, "SSL attempts to learn a rich yet easy-to-leverage feature space" which is exactly the purpose behind multi-fidelity models. In conclusion, I would like the authors to acknowledge that SSL is another way of creating models with "easy-to-leverage feature spaces" and there are prior works to do so using multi-fidelity models.
> >
> > 2. Given the results that the authors have shared about mixing the differential equations, I think the authors must tone down the claims about foundation models as there are more steps than just scaling the current framework. Nevertheless, this is certainly a step forward toward the goal. I am happy to see this!
> >
> > 3. While I appreciate the clarity of the authors in the response, I do think both SSL and neural operator-type methods would be needed. The example results of FNO and FNO + SSL look good. I would like to see something similar with DeepONets vs. DeepONets+SSL. In other words, a foundation model may not just be SSL, but we may also need ingredients of a neural operator to be incorporated. Therefore, I would like to see the SSL in conjunction with the neural operators.

---

> > > ### Author Response · Authors · 2023-08-15
> > > **Response to reviewer clarifications**
> > >
> > > Thank you for clarifying your question and adding further feedback. We have responded to the additional comments below.
> > >
> > > ***
> > >
> > > ### On Multi-Fidelity Modeling and General Discussion of Other Feature Learning Methods
> > >
> > > Thank you for pointing out the connection between multi-fidelity modeling and our approach using SSL. We appreciate your feedback and agree that recognizing related works is crucial for a holistic understanding of our work's context.
> > >
> > > We concur that both multi-fidelity modeling and SSL can reduce dependency on extensive training data and have practical implementations. Despite these similarities, the two methods serve somewhat different overall purposes, and exploit data in unique ways. Multi-fidelity modeling primarily combines data and models of varying fidelities. Common goals include training models “using data from different levels of fidelity” [1] or enhancing “accuracy by injecting a small set of high-fidelity observations” into less accurate models [2]. In contrast, SSL aims to harness salient features from diverse data sources without being tailored to specific applications. The techniques we employed capitalize on the inherent structure of our dataset, especially through augmentations and invariances.
> > >
> > > Looking more broadly, we acknowledge that SSL is not the sole approach to feature learning. There exists a myriad of techniques, including metric learning, kernel design, autoencoders, and others [3-4]. We opted for SSL due to its proven efficacy in fields like computer vision and the direct analogy offered by our data augmentations. Nonetheless, you are correct that there are some high-level commonalities between multi-fidelity modeling and our approach. We appreciate this observation, and will add multi-fidelity modeling to the discussion of related work in the final revision.
> > >
> > >
> > >
> > > ### On Foundation Models and Mixing Differential Equations
> > >
> > > Thank you for your appreciation of the new mixed equation experiment. As stated in our response, we will tone down the language surrounding foundation models and share more details of the mixed equation experiments in the final draft.
> > >
> > > ### On Experiments with DeepONets
> > >
> > > Thank you for emphasizing the potential synergy between SSL and various neural operator-type methods. Your insight on the possible complementarity between these approaches is well-taken.
> > >
> > > To your point on expanding our experiments: While we have showcased the benefits of integrating SSL with the FNO, a representative neural operator architecture, we understand the value of further extending this analysis to other architectures like DeepONets. In preparation for the final paper, we commit to providing experimental results with DeepONets in tandem with SSL. We would like to underscore, however, that while additional architectures will certainly offer a broader perspective, the underlying message of our work regarding the usefulness of SSL remains consistent.
> > > We resonate with your viewpoint that the strength of SSL is amplified when used in harmony with other state-of-the-art techniques, thereby underscoring the collaborative nature of innovation in this domain.
> > >
> > >
> > >
> > > **References:**
> > >
> > > [1] Fernández-Godino, M. Giselle, et al. "Review of multi-fidelity models." arXiv preprint arXiv:1609.07196 (2016).
> > >
> > > [2] Perdikaris, Paris, et al. "Nonlinear information fusion algorithms for data-efficient multi-fidelity modelling." Proceedings of the Royal Society A: Mathematical, Physical and Engineering Sciences 473.2198 (2017): 20160751.
> > >
> > > [3] Kaya, Mahmut, and Hasan Şakir Bilge. "Deep metric learning: A survey." Symmetry 11.9 (2019): 1066.
> > >
> > > [4] Williams, Christopher KI, and Carl Edward Rasmussen. Gaussian processes for machine learning. Vol. 2. No. 3. Cambridge, MA: MIT press, 2006.

---

> > > > ### Comment · Reviewer_swdq · 2023-08-17
> > > >
> > > > After reading the author's responses and other reviewers' responses, I have increased the score for this paper to accept; Cheers!

---

### Official Review · Reviewer_VTPv · 2023-07-12

**Soundness:** 4 excellent
**Presentation:** 4 excellent
**Contribution:** 2 fair
**Rating:** 5
**Confidence:** 4

**Summary:**

This paper presents a general framework for self-supervised learning in a PDE context. In a way that is principled and natural, PDE symmetry groups are used to make the requisite augmentations from which self-supervision will learn structure; the augmentations are selected carefully so as to keep the regressed quantity (for the downstream task) constant. The utility of this method is demonstrated by regressing quantities of interest on four downstream PDEs, and the results indicate that the improvement of predicting from self-supervised representations is considerable relative to straightforward supervised learning. This work paves the way for adapting self-supervision to the physical sciences by way of carefully considered symmetry group-oriented data augmentations.

**Strengths:**

1. The high-level idea is relatively novel, in that I do believe this is the first work that aims to make use of self-supervised learning for partial differential equations, with augmentations performed naturally according Lie symmetries. The most closely related work [12] uses Lie point symmetries of PDEs in order to augment PDE datasets, but this happens purely in the context of supervised tasks.

2. On the four PDE datasets presented/four equations considered, the results provide a fairly convincing improvement over just supervised learning alone. The mean squared error of prediction is considerably reduced (Table 1).

3. In terms of writing and presentation, the paper is quite good. The field is well introduced, with relevant literature cited. The idea is clearly presented and figures are given to illustrate the methodology (Figures 1, 2, and 3 were all quite instructive).

**Weaknesses:**

1. Although the idea of using Lie symmetry-based augmentation for self-supervised learning is novel, I was underwhelmed at the rigor with which these augmentations were applied. Symmetries of PDEs are with respect to infinite domain or periodic boundaries. The boundary conditions imposed in the paper violate such symmetries, and hence make the justification for this sort of operation theoretically dubious. The authors admit that, naturally, because they violate these symmetries, they can only implement the group operations with "small strengths." I believe that the work would benefit from further investigating how to preserve these boundary conditions during augmentation, thereby providing a theoretically sound basis for the self-supervised learning proposal.

2. One of the interesting discoveries of this paper is that learning on top of self-supervised representations is considerably better than just immediately employing supervised learning techniques (Table 1); this is in stark contrast to what we see in e.g. computer vision, where both exhibit roughly the same performance. Some cursory post hoc analysis is given to explain this observation, although no further investigation is done in terms of providing a good understanding (be it theoretical or empirical).

**Questions:**

My questions to the authors are listed below:

1. Have you put more thought into preserving boundary conditions during augmentation and/or rectifying the fact that the symmetry transformations applied are no longer the proper symmetries, since the system no longer has infinite domain?

2. Have you done any further investigation into why self-supervised learning gives results that considerably exceed that of supervised learning alone?

### Verdict

In this paper, the authors consider the novel idea of applying self-supervised learning to PDEs. Although they lay the basic groundwork (e.g. performing augmentation via Lie symmetries), I feel that more consideration is needed to find a way to preserve boundary conditions during augmentation, such that the work is theoretically principled. The results are promising, in that the proposed approach considerably outperforms supervised learning, but little investigation is done as to why (this is a surprising/interesting observation and I would like to see more of an explanation given). As such, I recommend a borderline reject rating for this paper.

### Additional Comments and Minor Corrections

The writing and presentation was, for the most part, very good (as mentioned above). I would recommend clarifying some of the table headings in results figures, for example, changing "Best strength" in Figure 5 (left) to "Best strength ($\epsilon$)." A few minor corrections are given below:

L221: "test samples.As shown in" -> "test samples. As shown in"

L229: "for to evaluate the models" -> "to evaluate the models"

L253: "buyoancy" -> "buoyancy"

L254: "which the hardest evaluation setting -> "which is the hardest evaluation setting"

## Post-rebuttal update

After reading the authors' rebuttals, overall, I think the paper will make for a positive contribution to the conference. I have increased my score to a 5.

**Limitations:**

The authors have adequately addressed limitations of their work in the "Discussion" section (Section 5).

---

> ### Author Rebuttal · Authors · 2023-08-10
>
> Thank you for taking the time to review our paper and for the valuable comments and suggestions. We are glad that you found our high-level idea novel and our work well-presented. We acknowledge the concerns raised and would like to address them as follows.
>
> ___
> ### On preserving boundary conditions during augmentation and/or rectifying the fact that the symmetry transformations applied are no longer the proper symmetries
>
> Your point about the use of Lie symmetries and their association with infinite domain or periodic boundaries is well-taken. As discussed in our draft, we agree that more rigorous treatment of this matter could fortify our approach. This is, in some ways, a rather fundamental challenge in SSL: a similar situation arises in the image setting, where symmetries such as resize or translation also incur boundary issues (for instance, if an important piece of the image is omitted by this augmentation).
>
> Importantly, our use of crops as an SSL augmentation mitigates this issue by biasing the network to learn local features that are in some sense robust to errors on the boundary. In essence, cropping ensures that learned features are invariant to whether the data was collected at a boundary or not. Both in the PDE and the image setting, this use of crops is crucial to the success of SSL.
>
> In more formally dealing with boundary conditions, we explored a number of ideas which are listed below. All of these ideas either significantly increased the complexity of the task or lost sight of the generality of the SSL PDE approach, so we opted to leave them to future work.
> - Approximate symmetries and discrete symmetries offer a way to explore many more symmetries, some of which may preserve boundary conditions. However, deriving these is no longer systematic: in the Lie symmetry case and in the approximate case, they provide solutions that are accurate only up to the order of the magnitude of the symmetry operation. These are also typically derived in settings with infinite domains. So apart from offering more flexibility in the symmetries, it is not obvious these directly help the boundary issue.
> - Another class of symmetries commonly studied are Lie Bäcklund symmetries [1], which offer a means to transform solutions from one PDE to another, but these are also typically derived in infinite domains.
> - Symmetries can be enforced at an infinitesimal level by using the Lie derivative defined as
> \begin{equation}\lim_{t\to 0} \frac{f\bigl(\Phi^t_X(p)\bigr) - f\bigl(p\bigr)}{t},\end{equation}
> where $\Phi^t_X$ is the exponential map for the Lie algebra vector field $X$ applied for an amount $t$. This Lie derivative can be enforced to be close to zero in the domain of the PDE outside of boundaries. Though this is nice in practice, the Lie derivative does not apply an augmentation, and we do not know how to apply this in SSL settings.
>
>
>
> To summarize, at least in this first work introducing the method, we wanted to offer a general approach to SSL for learning PDE data that was relatively simple to implement. The above approaches would have greatly complicated our approach, for questionable gain and loss of generality. As evidenced both in existing self-supervised work for images and our experiments, the representations learned even by these imperfect  augmentations still contain  rich information for downstream tasks. We welcome more thoughts and discussion from the reviewer on this point.
>
>
> ### Why SSL works better
>
> In general, understanding when and why SSL works is an active area of research. We know from prior theoretical and experimental results that, when done right, SSL pre-training finds feature spaces that are suited to diverse downstream targets and less prone to fitting trivial features [2-4]. From analyzing loss plots, we have some evidence that this is the case here.
>
> We have added loss plots for kinematic viscosity regression (Burgers') in Figure 1 (see attached PDF) explaining the observed gaps in Table 1 of our paper between “Supervised” and “SSL features”. Both methods rely on a ResNet18 architecture. In the “Supervised” setting, this network is trained from scratch, whereas for the “SSL features” method, the network was pre-trained with our SSL framework, subsequently frozen, and then a linear model was trained on top of the output features. “SSL features” (i) are rich yet can easily be leveraged with a linear layer, allowing fast convergence towards small test errors and (ii) are less prone to overfitting than supervised learning while using the same architecture. The gaps observed in Table 1 might be reduced via longer “Supervised” training or other efforts, but this would go out of the scope of a fair evaluation. SSL features are simply competitive and easier to use.
>
>
> **References:**
>
> [1] Ibragimov, Nail H. CRC handbook of Lie group analysis of differential equations. Vol. 3. CRC press, 1995.
>
> [2] Chen, Ting, et al. "A simple framework for contrastive learning of visual representations." International conference on machine learning. PMLR, 2020.
>
> [3] HaoChen, Jeff Z., et al. "Provable guarantees for self-supervised deep learning with spectral contrastive loss." Advances in Neural Information Processing Systems 34 (2021): 5000-5011.
>
> [4] Cabannes, Vivien, et al. "The ssl interplay: Augmentations, inductive bias, and generalization." International Conference on Machine Learning. PMLR, 2023.

---

> > ### Comment · Reviewer_VTPv · 2023-08-20
> > **Thank you for the rebuttal**
> >
> > I have read the rebuttal as well as the rebuttals to other authors' concerns. Cumulatively, I think the work lays a reasonable foundation and sets the stage for interesting follow-up work. I have increased my score to a 5.

---

### Author Rebuttal · Authors · 2023-08-10

## Summary Response

We thank the reviewers for their insightful comments and many great questions. We have responded to each reviewer’s comments separately, and are sharing a summary response covering the common threads in the reviewers’ responses. To enhance our responses, we have added experiments to back up many of the points we make below. We truly value further input from the reviewers on these matters.

***

### Replicating experiments for different architectures:
Multiple reviewers pointed out that they would be more confident in our results if we replicated time-stepping for different architectures, especially those that operate in Fourier space. We agree with this perspective, and are happy to share new results for different architectures:
- For Burgers: FNO1d, conditioned as detailed in our work.
- For Navier-Stokes: Fourier Neural Operator [2] with SpaSpec conditioning [3], Fourier U-Nets with Addition conditioning [3], and an additional conditioning method of U-Nets, AdaGN [3].




| Burger's (NMSE)               | ResNet1d     | FNO1d       |
|-----------------------------|--------------|-------------|
| No conditioning (baseline)  | 0.110 ± 0.008| 0.184 ± 0.002 |
| Representation conditioning | 0.108 ± 0.011| 0.173 ± 0.002 |



| Navier-Stokes (MSE x 1e-3)                                              | U-Nets + Addition | U-Nets + AdaGN | FNO + SpaSpec  | Fourier U-Nets + Addition |
|-----------------------------------------------------------------|-------------------|----------------|----------------|--------------------------|
| Time conditioning only (baseline)                                | 2.60 ± 0.05       | 2.37 ± 0.01    | 13.4 ± 0.5    | 3.31 ± 0.06             |
| Time conditioning + representation conditioning (ours)           | 2.47 ± 0.02       | 2.35 ± 0.03    | 13 ± 1.0        | 2.37 ± 0.05             |




### Handling boundary conditions:

As two reviewers have pointed out and as detailed in the limitations section of our work, boundary conditions can be a source of error when implementing Lie point symmetries. Since virtually all symmetry derivations are done assuming infinite or periodic domains, it is not necessarily obvious how to mathematically handle these boundaries in derivations of symmetries. In practice, this technical issue is not as major as one would expect:
- A similar situation arises in image settings where augmentations like resize or translation cause issues on the boundary. However, the goal is to learn global features that are invariant to these boundary issues. In pursuit of that higher level goal, SSL is still effective at learning with these augmentations included.
- Crops are an essential augmentation both here and in the standard image setting [1]. For PDEs, this is partly because crops help bias the network to learn features that are invariant to whether the input was taken near a boundary or not.
- There is no obvious technical solution to this problem as far as we are aware. Indeed, other forms of PDE symmetry groups like approximate symmetries, discrete symmetries, or Lie Bäcklund symmetries are also derived in infinite domains and do not address boundary issues. These symmetry groups are typically much harder to derive, more complicated, and do not necessarily come with the nice Lie algebraic structure associated with the Lie point symmetry group.

There is much more to say on this point and our more detailed thoughts are captured in our response to reviewer VTPv which we recommend that reviewers look at. We are open to more ideas and happy to continue the discussion on this point.



**References:**

[1] Chen, Ting, et al. "A simple framework for contrastive learning of visual representations." International conference on machine learning. PMLR, 2020.

[2] Li, Zongyi, et al. "Fourier neural operator for parametric partial differential equations." arXiv preprint arXiv:2010.08895 (2020).

[3] Gupta, Jayesh K., and Johannes Brandstetter. "Towards multi-spatiotemporal-scale generalized pde modeling." arXiv preprint arXiv:2209.15616 (2022).

---

### Decision · Program_Chairs · 2023-09-21

**Decision:**

Accept (poster)

**Comment:**

The paper proposes to use self-supervised learning for improving certain PDE-related problems. Many PDEs come with point Lie symmetries, so this can generate different 'views' of the dataset, thus standard techniques can be used.
The numerical examples report significant improvement over baselines in some cases.
The task considered in the paper is regression of the coefficients, i.e. efficient representation is indeed important.

The reviewers agree that the paper is well-written. However, the problem itself is not very common (regression task) and is indeed a classical regression task. Thus, the paper combines two well-known facts: some PDE has symmetries;
this can be used together with SSL, and there is no much specifics about that. Also, the question of boundary condition is valid, thus making the method even more empirical.

Overall, although the paper is a combination of two standard techniques, it has not been shown to be effective before in this particular domain (inverse problems).The more detailed studies will follow.